# Profiling ubiquitin signalling with UBIMAX reveals DNA damage- and SCF$^{\beta\text{-Trcp1}}$-dependent ubiquitylation of the actin-organizing protein Dbn1

Camilla S. Colding-Christensen [1,4], Ellen S. Kakulidis[1,4], Javier Arroyo-Gomez [1], Ivo A. Hendriks [1], Connor Arkinson[2,3], Zita Fábián [1], Agnieszka Gambus [2], Niels Mailand [1], Julien P. Duxin [1,5] ✉ & Michael L. Nielsen [1,5] ✉

Ubiquitin widely modifies proteins, thereby regulating most cellular functions. The complexity of ubiquitin signalling necessitates unbiased methods enabling global detection of dynamic protein ubiquitylation. Here, we describe UBIMAX (UBiquitin target Identification by Mass spectrometry in *Xenopus* egg extracts), which enriches ubiquitin-conjugated proteins and quantifies regulation of protein ubiquitylation under precise and adaptable conditions. We benchmark UBIMAX by investigating DNA double-strand break-responsive ubiquitylation events, identifying previously known targets and revealing the actin-organizing protein Dbn1 as a major target of DNA damage-induced ubiquitylation. We find that Dbn1 is targeted for proteasomal degradation by the SCF$^{\beta\text{-Trcp1}}$ ubiquitin ligase, in a conserved mechanism driven by ATM-mediated phosphorylation of a previously uncharacterized β-Trcp1 degron containing an SQ motif. We further show that this degron is sufficient to induce DNA damage-dependent protein degradation of a model substrate. Collectively, we demonstrate UBIMAX's ability to identify targets of stimulus-regulated ubiquitylation and reveal an SCF$^{\beta\text{-Trcp1}}$-mediated ubiquitylation mechanism controlled directly by the apical DNA damage response kinases.

Ubiquitin is a small 76 amino acid protein, which can be attached via its C-terminus to target proteins via the catalysis of specific ubiquitin conjugating enzymes[1–3]. It is estimated that this dynamic post-translational modification (PTM), ubiquitylation, regulates nearly all cellular functions[4]. As ubiquitin can be attached to target proteins as monomers and as chains of different topologies, the resulting signal can be highly complex. These signals are decoded by proteins with the ability to interact with specific ubiquitin topologies. Such proteins are often effectors of cellular pathways, allowing ubiquitylation topologies to regulate distinct cellular functions[4,5]. For instance, many aspects of the DNA damage response (DDR) are regulated by K63- and K48-ubiquitin signalling, including recruitment of effectors to the DNA damage site and choice of repair pathway[6]. One example is the DNA double-strand break (DSB)-induced K48-linked ubiquitylation of the Ku complex, which causes its eviction from DNA, thus regulating the DSB repair process[7–9]. While extensive biochemical and molecular

[1]The Novo Nordisk Foundation Center for Protein Research, Faculty of Health and Medical Sciences, University of Copenhagen, 2200 Copenhagen, Denmark. [2]Institute of Cancer and Genomic Sciences, University of Birmingham, Birmingham, UK. [3]Present address: California Institute for Quantitative Biosciences and Department of Molecular and Cell Biology and Howard Hughes Medical Institute, University of California at Berkeley, Berkeley, CA 94720, USA. [4]These authors contributed equally: Camilla S. Colding-Christensen, Ellen S. Kakulidis. [5]These authors jointly supervised this work: Julien P. Duxin, Michael L. Nielsen. ✉e-mail: julien.duxin@cpr.ku.dk; michael.lund.nielsen@cpr.ku.dk

analyses have been required for such investigations, the breadth and complexity of ubiquitin signalling has prompted the need for unbiased and global ubiquitin detection methods.

Mass spectrometry (MS)-based proteomics has emerged as a valuable tool for studying the ubiquitin landscape and has been widely used to investigate ubiquitylation events in the DDR and DNA repair processes[10]. Over the last decade, numerous MS-based approaches have been established to identify ubiquitylated proteins, determine the amino acid acceptor sites, establish the chain-topology of ubiquitylation, and identify the enzymes involved[10–12]. Particularly the methods for identifying ubiquitylation sites have been extensively used for profiling ubiquitylation responses[13–17]. Although these methods have provided valuable insights into the ubiquitin landscape, they are limited in providing quantitative information of the ubiquitylated target. While methods for identification of ubiquitylation on the protein level can provide such quantitative information about ubiquitylated proteoforms[18–22], these methods are often challenged by a lack of specificity in detecting ubiquitin-conjugated *versus* -interacting proteins. Furthermore, current methods have limitations when it comes to capturing steady-state systems and targeting specific events in response to a particular stimulus. For instance, it is challenging to generate site-specific DNA lesions in cellular model systems, prompting the need for developing an in vitro system to precisely study ubiquitylation responses to defined stimuli.

The *Xenopus* egg extract model system has been extensively used for biochemical and molecular studies of DNA metabolism and genome maintenance mechanisms[23–25]. This is chiefly owing to the possibility of investigating key biological processes with high spatiotemporal resolution in the absence of essential proteins, and upon the ready addition of recombinant proteins or specific inhibitors. Recently, *Xenopus* egg extracts have been coupled to MS-based proteomic analyses to study quantifiable changes in protein recruitment to damaged DNA[26,27] as well as for identifying small ubiquitin-like modifier substrates through a tagged protein approach[28].

Here, we describe an MS-based method, referred to as UBIMAX (UBiquitin target Identification by Mass spectrometry in Xenopus egg extracts), which allows for detection of global, specific, and quantifiable changes in de novo protein ubiquitylation under precise and modifiable biological conditions in *Xenopus* egg extracts. As proof of principle, we use UBIMAX to identify previously characterized DNA damage-induced ubiquitylation events alongside several previously uncharacterized targets of DNA damage-specific ubiquitylation in response to DSBs and DNA-protein crosslinks (DPCs). From this, we discover that the actin-organizing protein Dbn1, which has not been previously linked to the DDR, is a prominent ubiquitylation target in response to DSBs. Using *Xenopus* egg extract and human cells, we demonstrate that DSB-induced Dbn1 ubiquitylation depends on the apical DDR kinase ATM, is mediated by the Skp1-Cul1-F-box$^{\beta\text{-Trcp1}}$ (SCF$^{\beta\text{-Trcp1}}$) complex and leads to proteasomal degradation of Dbn1. Additionally, we uncover a DNA damage-responsive β-Trcp1 degron in the C-terminal unstructured region of Dbn1 and show that it is necessary and sufficient for conferring DSB-induced ubiquitylation and degradation of a target protein. Collectively, we demonstrate the capability of UBIMAX to identify players in ubiquitylation responses under specific biological conditions of interest. Furthermore, by deciphering the mechanism of DSB-induced Dbn1 ubiquitylation identified by UBIMAX, we reveal a variant β-Trcp1 degron that mediates a DDR- and SCF$^{\beta\text{-Trcp1}}$-specific ubiquitylation and degradation programme.

## Results

### UBIMAX specifically detects ubiquitin-conjugated proteins

To establish a method for identification of endogenous ubiquitylation events in response to specific stimuli, we combined high-resolution mass spectrometry (MS) with the malleable *Xenopus* egg extract system[25]. Briefly, we supplemented high speed supernatant interphase egg extracts (HSS) with recombinant 6xHis-tagged ubiquitin prior to inducing a stimulus-driven response (Fig. 1a). Following a specific stimulation, endogenous and supplemented ubiquitin was allowed time to conjugate onto target proteins, enabling highly stringent enrichment of target proteins via pulldown of His$_6$-ubiquitin. Next, the enriched ubiquitylated proteins were digested on-beads using trypsin, with the resulting peptides purified and concentrated on C$_{18}$-StageTips[29] and followed by their characterization via label-free quantitative MS[30].

To minimize non-specific ubiquitylation events, we ensured that recombinant 6xHis-tagged ubiquitin was titrated to equimolar amounts of endogenous ubiquitin (Fig. 1b). Under these conditions, recombinant 6xHis-tagged ubiquitin was efficiently conjugated onto target proteins (Fig. 1c lanes 3-8 and Supplementary Fig. 1a), thus allowing for efficient enrichment of these proteins using the UBIMAX approach.

As ubiquitylation constitutes an important part of the response to DNA damage[6], we chose DSBs as our stimulus of choice for benchmarking UBIMAX. To elicit a DSB response, we added linearized plasmid DNA to egg extracts, while omission of DNA or addition of circular undamaged plasmid DNA served as controls. Importantly, addition of recombinant 6xHis-tagged ubiquitin did not activate the DDR in the absence of DNA (Fig. 1c, lanes 3-4), nor did it affect DDR activation by the linearized plasmid, as indicated by Chk1-S345 phosphorylation (Fig. 1c, lane 8). DSB repair was also unaffected by addition of recombinant 6xHis-tagged ubiquitin, and linearized plasmids were ligated by non-homologous end joining (NHEJ) irrespectively of the presence or absence of recombinant ubiquitin[31–33] (Supplementary Fig. 1b and quantifications in Supplementary Fig. 1c). In contrast, NHEJ-mediated repair of DSBs was partially impaired in the presence of a ubiquitin E1 inhibitor[34], confirming the relevance of de novo ubiquitylation for DSB repair in egg extracts (Supplementary Fig. 1b, c). Collectively, we established appropriate conditions to study protein ubiquitylation in response to DSBs.

Next, we sought to profile DSB-induced ubiquitylation events using UBIMAX. To ensure that the method enriches ubiquitin-conjugated target proteins, as opposed to ubiquitin-interacting proteins, we performed all enrichments under denaturing conditions. However, as background binding proteins are an inherent challenge in any enrichment-based proteomic experiment[35], we utilized label-free quantification (LFQ)[30] of replicate samples to be able to distinguish background binding proteins from true ubiquitin target proteins. To distinguish between specific and non-specific enrichment of proteins, we additionally performed reactions in egg extracts supplemented with either recombinant 6xHis-tagged or untagged ubiquitin (Fig. 1d, Supplementary Data 1a). As an additional control, we performed reactions in the presence or absence of ubiquitin E1 inhibitor to block ubiquitylation. Finally, as described above, we investigated DSB-induced ubiquitylation events by adding linearized plasmid DNA, circular plasmid DNA, or no plasmid DNA to individual reactions. We performed all reactions in quadruplicate and collected samples 30 min after addition of DNA. Samples were subsequently subjected to the UBIMAX workflow (Fig. 1a). Overall, we observed very high reproducibility across all replicates and sample groups ($R = 0.94–0.99$) (Fig. 1e) with low within-sample group median coefficients of variation (Supplementary Fig. 1d), supporting that the dynamics of the investigated ubiquitylation landscape were highly specific. This was corroborated by principal component analysis (PCA), which revealed a large variation between controls and ubiquitin target enriched sample groups (Supplementary Fig. 1e). Correspondingly, UBIMAX robustly separated the ubiquitin target enriched sample groups based on the introduced DNA stimuli (Supplementary Fig. 1f).

To assess the efficiency of our enrichment approach, we analysed the average contribution of ubiquitin peptides to total sample signal (Supplementary Fig. 1g) and found it significantly higher in ubiquitin

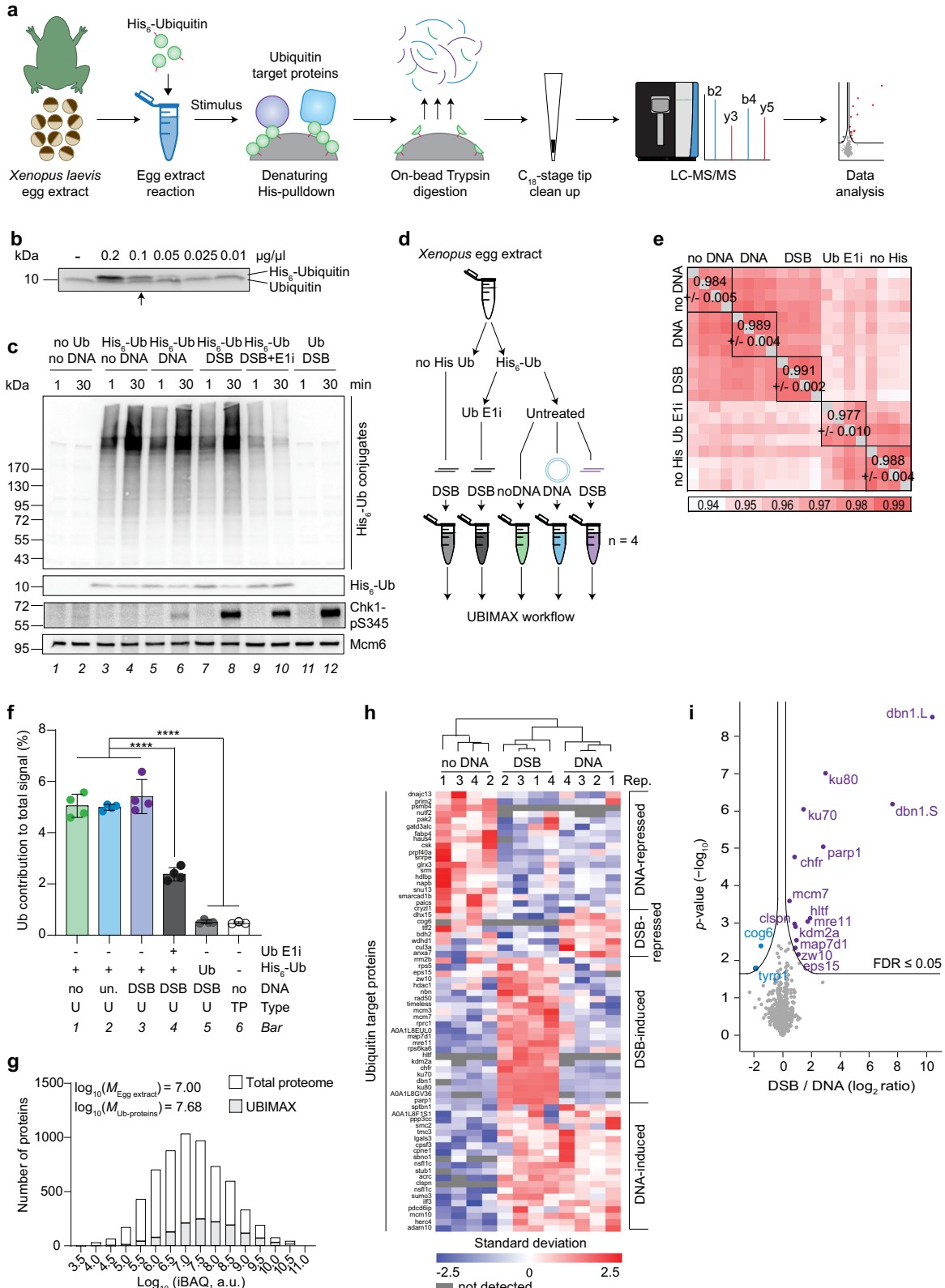

target enriched sample groups compared to each of the controls (Fig. 1f, compare bars 1-3 with 4 and 5). Importantly, the ubiquitin contribution in the untagged ubiquitin control was similar to that of a total proteome (Fig. 1f, compare bars 5 and 6, and Supplementary Data 2). This shows that our denaturing His₆-ubiquitin enrichment approach is highly efficient with ~90% of the ubiquitin signal being specific (Fig. 1f, ratio of bars 1-3 and 5). Furthermore, ubiquitin

contribution in the ubiquitin target enriched sample groups was approximately two-fold higher than in the ubiquitin E1 inhibitor control (Fig. 1f, compare bars 1-3 with 4). As a large fraction of recombinant 6xHis-tagged ubiquitin is left unconjugated in the presence of the ubiquitin E1 inhibitor (Fig. 1c, lanes 9-10), this indicates that while non-conjugated His₆-ubiquitin is efficiently recovered in the presence of ubiquitin E1 inhibitor, a similar amount of endogenous ubiquitin is

**Fig. 1 | UBIMAX efficiently and specifically detects ubiquitin-conjugated proteins in response to DSBs. a** Schematic representation of UBIMAX experimental system and workflow. LC-MS/MS; liquid chromatography-tandem mass spectrometry. **b** $His_6$-ubiquitin was added to extracts at indicated concentrations and analysed by western blot. Arrow indicates concentration of $His_6$-ubiquitin, 0.1 μg/μL, used in subsequent experiments. **c**, **d** Extracts were supplemented with DMSO or ubiquitin E1 inhibitor ("E1i") prior to addition of untagged ubiquitin ("Ub") or $His_6$-Ubiquitin ("$His_6$-Ub"). Reactions were initiated by addition of buffer ("no DNA"), undamaged plasmid DNA ("DNA") or linearized plasmid DNA ("DSB"). Samples were analysed by western blot (**c**) 1 or 30 min after reaction initiation. UBIMAX (experimental outline in **d**) was performed in independent reaction quadruplicates. Samples were harvested at 30 min and subjected to the UBIMAX workflow outlined in a. no His Ub, untagged ubiquitin; Ub E1i, ubiquitin E1 inhibitor. **e** Pearson correlation matrix of the experiment outlined in d. Within-replicate mean and ±standard deviation is indicated. no His, untagged ubiquitin. **f** Mean percent contribution of ubiquitin peptides to summed peptide abundance for UBIMAX samples ("U"), derived from quadruplicate independent reactions, and three biologically independent replicates of a total extract proteome ("TP"). Error bars represent standard deviations. Significance was determined by one-way ANOVA with Tukey's multiple comparisons test for all pairwise comparisons, with those indicated of $p$-values ' 0.0001. no, no DNA; un., undamaged plasmid DNA. **g** Depth of sequencing illustrated as distribution of $\log_{10}$(iBAQ)-values of ubiquitylated proteins detected by UBIMAX compared to proteins detected in a total extract proteome. Frequency distribution medians ("$M$") are shown at the top left corner. iBAQ, intensity-based absolute quantification; a.u., arbitrary units. **h** Hierarchical clustering analysis of Z-scored ubiquitylated protein abundances robustly changing with DNA treatment. Gene names are provided on the left (or UniProtID if unannotated). Rep., replicate. **i** Volcano plot analysis comparing ubiquitylated proteins enriched from DSB- *versus* undamaged DNA-treated samples. Purple and blue dots indicate significantly enriched and -depleted ubiquitylated proteins. Significance was determined by two-tailed Student's $t$-test, with permutation-based FDR-control with s0 = 0.1 and 2500 rounds of randomization, to ensure FDR ≤ 0.05. Source data are provided as a Source Data file.

additionally recovered in the stimulus groups. This observation is presumably due to the equal concentration of endogenous and recombinant 6xHis-tagged ubiquitin (Fig. 1b) and suggest that these are equally conjugated onto target proteins.

Finally, as ubiquitylation is a sub-stoichiometric PTM, we assessed the abundance bias of UBIMAX. Overall, we observed a large dynamic range spanning seven orders of magnitude ($3.76 \times 10^7$-fold) with a distribution of ubiquitylated proteins identified by UBIMAX similar to that of a total egg extract proteome ($\log_{10} M = 7.68$ and 7.00, respectively) (Fig. 1g and Supplementary Data 2), indicating a relatively modest abundance bias for UBIMAX. Collectively, these data show that UBIMAX is an efficient and robust method for identifying specific ubiquitylation events in *Xenopus* egg extracts.

## UBIMAX identifies DSB-induced protein ubiquitylation

From quadruplicate experiments, UBIMAX identified 1526 proteins of which 786 were significantly enriched by the His-ubiquitin pull-down and de novo ubiquitylated across the ubiquitin target enriched sample groups (Supplementary Fig. 1h, left and Supplementary Data 1a). We further interrogated a subset of these 786 proteins whose ubiquitylation status was consistently up- or downregulated across replicates and identified four clusters of specifically regulated ubiquitylated proteins in response to the DNA treatments (Fig. 1h). Enrichment analysis of the "DSB-induced" cluster revealed that ubiquitylation of proteins involved in DNA repair and DNA replication was upregulated in response to DSBs (Fig. 1h, Supplementary Fig. 1i and Supplementary Data 1b). This cluster included well-known DDR proteins such as the Ku70-Ku80 dimer, the Mre11-Rad50-Nbs1 (MRN) complex, and Parp1 (Supplementary Fig. 1j). It also included DNA replication factors such as Mcm3, Mcm7, and Timeless.

Next, we interrogated the 39 proteins which showed a significant regulation of ubiquitylation status upon stimulation with either undamaged or DSB-containing plasmid DNA (Supplementary Fig. 1h, right and Supplementary Data 1a). Volcano plot analysis of these ubiquitylation events confirmed the previously described DSB-induced ubiquitylation of Ku80, Parp1, Mre11, and Claspin[7,36–41] (Fig. 1i). Moreover, we also detected enrichment of ubiquitylated Hltf and Chfr, two ubiquitin E3 ligases known to auto-ubiquitylate upon DNA damage[38,42,43]. Finally, we detected DSB-induced ubiquitylation of proteins not previously described as being modified upon DSBs, including Mcm7. Strikingly, the most prominently induced ubiquitylated protein detected in response to DSBs was the actin-organizing protein Dbn1, a protein not previously connected with the DSB response. In conclusion, we demonstrate the ability of UBIMAX to reveal regulation of protein ubiquitylation events in response to DNA damage.

## UBIMAX identifies DNA damage specific ubiquitylation events

To further assess the capability of UBIMAX to detect ubiquitylation events triggered by a specific stimulus, we used UBIMAX to analyse the ubiquitylation response to different DNA lesions. To this end, we used plasmids carrying either the previously described *Haemophilus parainfluenzae* methyltransferase M.HpaII crosslinked at a single-stranded DNA gap ("ssDNA-DPC")[26], or the *Saccharomyces cerevisiae* recombinase Flp crosslinked at a single-strand break ("SSB-DPC")[44] (Fig. 2a). Repair of these DPC substrates have previously been shown to require ubiquitylation[26,45,46], thus making these DNA lesions relevant for UBIMAX analysis. Furthermore, while both are DPC lesions, the nature of the protein adducts and the DNA context (located on ssDNA *versus* at a SSB) are different, thus serving as a suitable test for the specificity of UBIMAX in distinguishing the ubiquitylation response to these lesions.

For profiling ubiquitylation events by UBIMAX in response to each of the DPC-containing plasmids, we also included the ubiquitin E1 inhibitor control. Reactions were performed in triplicate with samples collected 30 min after addition of DNA and subjected to the UBIMAX workflow (Fig. 1a; Supplementary Fig. 2a). From this, we detected distinct de novo protein ubiquitylation events induced by either ssDNA-DPC or SSB-DPC (Fig. 2b-c and Supplementary Data 3). While some proteins were found ubiquitylated in response to both substrates (e.g. Chfr, and Rpa1), UBIMAX also detected proteins uniquely ubiquitylated in response to either plasmid (e.g. Aplf for ssDNA-DPC; HelB for SSB-DPC). Next, we compared the proteins showing upregulation of ubiquitylation in response to the different DNA lesions (DSB vs ssDNA-DPC vs SSB-DPC) as detected by UBIMAX (Supplementary Fig. 2b). From this, we found that each type of DNA damage predominantly induced DNA damage-specific ubiquitylation events, with a few factors ubiquitylated in more than one condition. Consistent with their role in NHEJ and DSB repair, Ku80, Ku70, and Mre11 were specifically ubiquitylated in the presence of the DSB containing plasmid (Fig. 2d-f). In contrast, ubiquitylation of the ssDNA binding protein RPA[47] was greatly stimulated by the DPC lesions flanked by either ssDNA or a SSB, consistent with RPA binding to these substrates (in the case of the SSB-DPC, presumably once the SSB has been resected) (Fig. 2g). The only protein ubiquitylated in response to all three DNA lesions was Chfr (Fig. 2h), which is recruited to DNA damage sites in a poly(ADP-ribose) (PAR)-dependent manner[38], with PARylation likely occurring at all of these DNA lesions. Finally, we found that the actin-organizing protein Dbn1 was ubiquitylated in response to DSBs and, to a lesser degree, DPCs flanked by a ssDNA gap but not in response to DPCs flanked by a SSB (Fig. 2i).

Consistent with the UBIMAX data, western blot analysis confirmed that Dbn1 and Ku80 were ubiquitylated primarily in response to DSBs and to a much lesser extent in response to ssDNA-DPCs (Fig. 2j, lanes 4-6 and 10-12). In conclusion, we demonstrate the precision of UBIMAX

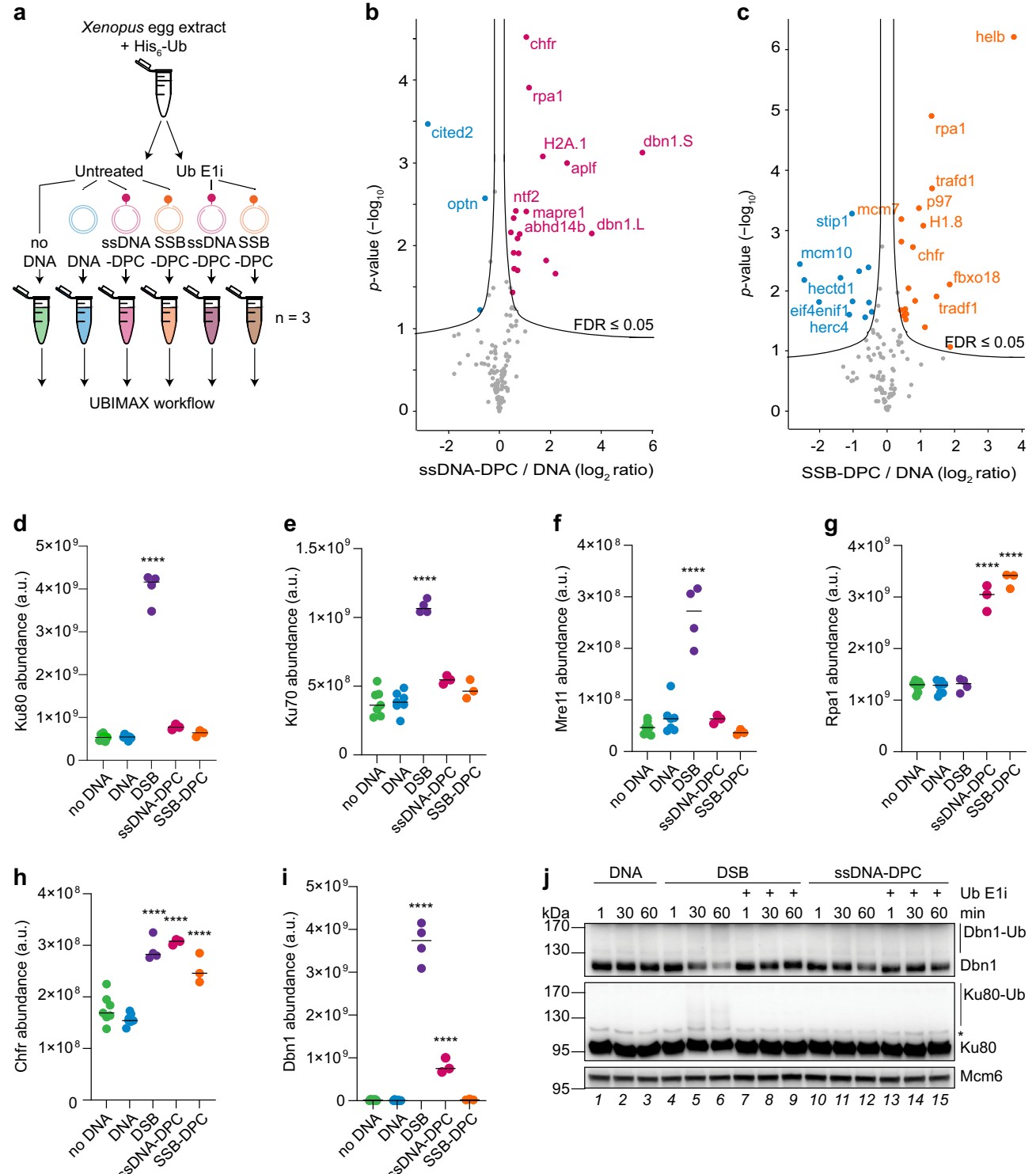

in detecting DNA damage specific de novo protein ubiquitylation events and identify the DNA damage-induced ubiquitylation of the actin-organizing protein Dbn1 primarily in response to DSBs.

## Ubiquitylation of Dbn1 results in proteasomal degradation

To validate the ability of UBIMAX to identify previously uncharacterized ubiquitylated substrates, we sought to further investigate the previously unknown, damage-induced ubiquitylation of Dbn1. To this end, we first generated a Dbn1 antibody that efficiently immunodepleted Dbn1 from egg extracts (Supplementary Fig. 3a). We then performed denaturing His-ubiquitin pulldowns from mock- or Dbn1

immunodepleted egg extracts supplemented with 6xHis-tagged ubiquitin and DSB plasmid and analysed the recovered proteins by western blot (Fig. 3a). The Dbn1 signal recovered from mock immunodepleted samples migrated as a smear 30 min after addition of DSBs and peaked at 60 min (Fig. 3a, lanes 1-3). In contrast, no signal was detected in Dbn1 immunodepleted extracts (Fig. 3a, lanes 4-6), validating the prominent poly-ubiquitylation of Dbn1 following DSBs.

We next investigated the mechanism of DNA damage induced Dbn1 ubiquitylation. We first addressed whether Dbn1 ubiquitylation required DDR activation. To investigate this, egg extract reactions were performed in the absence or presence of specific inhibitors of the

**Fig. 2 | UBIMAX identifies DNA damage specific ubiquitylation events and detects DSB-induced ubiquitylation of Dbn1. a** Experimental outline for the UBIMAX experiment profiling ubiquitylated proteins in response to DPC-containing substrates. Extracts were untreated or supplemented with ubiquitin E1 inhibitor ("Ub E1i") prior to addition of His$_6$-Ubiquitin ("His$_6$-Ub"). Reactions were initiated by addition of buffer ("no DNA"), undamaged plasmid DNA ("DNA"), plasmids carrying the M.HpaII protein crosslinked at a single-stranded DNA gap ("ssDNA-DPC"), or plasmids carrying the Flp protein crosslinked at a single-strand break ("SSB-DPC"). Reactions were performed in independent reaction triplicates. Samples were harvested at 30 min and subjected to the UBIMAX workflow outlined in Fig. 1a. **b, c** Volcano plot analysis comparing ubiquitylated proteins enriched from ssDNA-DPC (**b**) or SSB-DPC (**c**) *versus* DNA-treated samples. Pink/orange and blue dots indicate significantly enriched and -depleted ubiquitylated proteins. Significance was determined by two-tailed Student's *t*-test, with permutation-based FDR-control with s0 = 0.1 and 2500 rounds of randomization, to ensure an FDR ≤ 0.05. Ubiquitylated proteins with FDR ≤ 0.01 are labelled. **d–i** Abundance of Ku80 (**d**) Ku70 (**e**) Mre11 (**f**) Rpa1 (**g**) Chfr (**h**) and Dbn1 (**i**) across ubiquitin target enriched samples of the UBIMAX experiments profiling protein ubiquitylation in response to DSBs (Fig. 1d) and DPCs (a). Horizontal lines indicate the median and significance was determined by one-way ANOVA with Dunnett's multiple comparisons test for all conditions against undamaged DNA with a cut-off of *p*-value ≤ 0.01 and with those indicated of *p*-values < 0.0001. *n* = 3 independent reaction replicates for DSB and DPC conditions and *n* = 7 for no DNA and DNA conditions. a.u., arbitrary units. **j** Extracts were untreated or supplemented with ubiquitin E1 inhibitor prior to initiation of reactions by addition of undamaged plasmid DNA ("DNA"), linearized plasmid DNA ("DSB"), or plasmids carrying a DPC at a ssDNA gap ("ssDNA-DPC"). Samples were analysed by western blot at the indicated times. *unspecific band. Source data are provided as a Source Data file.

apical DDR kinases, ATM and ATR, prior to stimulation with DSB plasmid DNA (Fig. 3b and Supplementary Fig. 3b). Addition of DSB plasmid DNA to egg extracts activated the DDR within minutes, as evidenced by the appearance of Chk1-S345 phosphorylation, followed by the appearance of Dbn1 ubiquitylation at 60 min (Fig. 3b, lanes 1-5). Inhibition of either ATM or ATR inhibited DDR activation and DSB-induced Dbn1 ubiquitylation (Fig. 3b lanes 6-10 and Supplementary Fig. 3b lanes 7-18), suggesting that Dbn1 is ubiquitylated in response to DDR kinase activation.

To investigate the consequences of Dbn1 ubiquitylation, we next examined the major ubiquitin chain topologies present on the protein, as chain topology directs the functional consequences of protein ubiquitylation[4,5]. Particularly, K63- and K48-linked ubiquitin chains have been shown to orchestrate the response to DSBs[6]. To this end, we assessed the extent of Dbn1 ubiquitylation in the presence of DSB plasmid DNA in *Xenopus* egg extracts supplemented either with an excess of recombinant 6xHis-tagged wild-type (WT) ubiquitin or various chain deficient mutants (Supplementary Fig. 3c). In the presence of WT ubiquitin, Dbn1 was heavily poly-ubiquitylated and migrated as different high molecular weight species on the gel (Supplementary Fig. 3c lane 1). In contrast, addition of a ubiquitin mutant unable to form lysine-linked chains (i.e. all lysines substituted with arginines, referred to as "noK") resulted in faster migrating Dbn1 species consistent with conjugation of shorter ubiquitin chains or multiple mono-ubiquitylation (Supplementary Fig. 3c lane 6). The high molecular weight species of Dbn1 were maintained in the presence of ubiquitin variants either unable to form K63-linked chains ("K63R") or ubiquitin only capable of forming K48-linked chains ("K48only") (Supplementary Fig. 3c lanes 3-4). Conversely, upon addition of a ubiquitin variant unable to form K48-linked chains ("K48R") or ubiquitin able to form K63-linked chains only ("K63only"), the ubiquitylated Dbn1 species migrated faster, similar to that observed with the noK ubiquitin mutant (Supplementary Fig. 3c lanes 2, 5 and 6). Collectively, these data support that DSBs mainly induce K48-linked poly-ubiquitylation of Dbn1.

As K48-linked poly-ubiquitylation is a canonical signal for proteasomal degradation of the targeted protein[4,5,48], we next examined whether DSB-induced ubiquitylation of Dbn1 would target the protein for degradation. Indeed, addition of proteasome inhibitor to egg extracts greatly stabilized ubiquitylated Dbn1 in the presence of DSB plasmid DNA (Supplementary Fig. 3d). Overall, we conclude that DSB-induced activation of the ATM/ATR-mediated DDR elicits K48-linked poly-ubiquitylation of the actin-organizing protein Dbn1, resulting in its proteasomal degradation.

### DSB-induced ubiquitylation of Dbn1 is mediated by SCF$^{β-Trcp1}$

To elucidate the mechanism of DSB-induced Dbn1 ubiquitylation, we aimed at identifying the ubiquitin E3 ligase responsible for the modification. The largest family of ubiquitin E3 ligases are the Cullin-RING ligases (CRLs), which concurrently are known to primarily induce K48-linked poly-ubiquitylation of substrates resulting in their proteasomal degradation[49,50]. We therefore reasoned that DSB-induced ubiquitylation of Dbn1 could be mediated by a CRL complex. To test this, we supplemented egg extracts with a pan-Cullin inhibitor ("Culi") prior to induction of the DDR by DSB plasmid DNA and found that the inhibitor abolished DSB-induced ubiquitylation and stabilized the Dbn1 protein (Fig. 3c).

In our UBIMAX analyses, we noted that ubiquitylation of Dbn1 followed a similar induction as ubiquitylation of Ku80 in response to DSBs (Figs. 1i, 2d and 2i). Ku80 is known to be ubiquitylated by the Skp1-Cul1-Fbxl12 (SCF$^{Fbxl12}$) complex while located on DSB DNA, triggering the dissociation of the Ku-complex from DNA[7,41]. Consequently, we wondered whether the two proteins also shared the Cullin-dependent mechanism of ubiquitylation. To explore this, we supplemented egg extracts with recombinant dominant negative Cul1, Cul3, Cul4a and/or -b or Cul5 protein prior to stimulation with DSB plasmid DNA (Fig. 3d). As expected, only dominant negative Cul1 abolished Ku80 ubiquitylation and stabilized the unmodified protein (Fig. 3d lanes 4-6). Similar to Ku80, we observed loss of Dbn1 ubiquitylation and corresponding stabilization of the unmodified Dbn1 protein only in the presence of dominant negative Cul1 or upon inhibition of all Cullin E3 ligases using an inhibitor of neddylation (Fig. 3d lanes 4-6 and 22-24). While we cannot exclude a potential contribution of other ubiquitin ligases not assayed for, we further validated Cul1-dependent ubiquitylation of Dbn1 by immunodepletion of Cul1 from egg extracts. This abolished ubiquitylation and completely stabilized both Ku80 and Dbn1 in the presence of DSB plasmid DNA (Supplementary Fig. 3e compare lanes 1-4 with 5-8).

Having established that Dbn1 is targeted for ubiquitylation in a Cul1-dependent manner we next sought to identify the Cul1 substrate targeting protein responsible for Dbn1 ubiquitylation. In addition to Cul1, Cul1 E3 ligase complexes consist of the RING-containing protein Rbx1, adaptor protein Skp1 and a substrate targeting F-box protein[49,51,52]. Since Dbn1 and Ku80 share the SCF-mediated mechanism of ubiquitylation upon DSBs, we first tested whether Dbn1 ubiquitylation also depended on the F-box protein Fbxl12. However, immunodepletion of Fbxl12 in egg extracts only caused stabilization of unmodified Ku80 but had no effect on DSB-induced ubiquitylation of Dbn1 (Supplementary Fig. 3e lanes 9-12). Furthermore, while we readily detected ubiquitylation of Ku80 on DSB DNA, we did not detect Dbn1 localization to DSB DNA (Supplementary Fig. 3f). This suggests that the SCF complex utilizes different F-box proteins to recognize and target differently localized Ku80 and Dbn1 for DSB-induced ubiquitylation, respectively.

Considering that the SCF complex can interact with >70 F-box proteins[53], we took advantage of a mass spectrometry-based approach to explore the mechanism for recognition and ubiquitylation of Dbn1 in response to DSBs (Fig. 3e). As we found that DSB-induced ubiquitylation of Dbn1 required ATM activity (Fig. 3b), we reasoned that a DSB-induced interaction between Dbn1 and the SCF complex would depend on ATM activity. Therefore, we performed a Dbn1

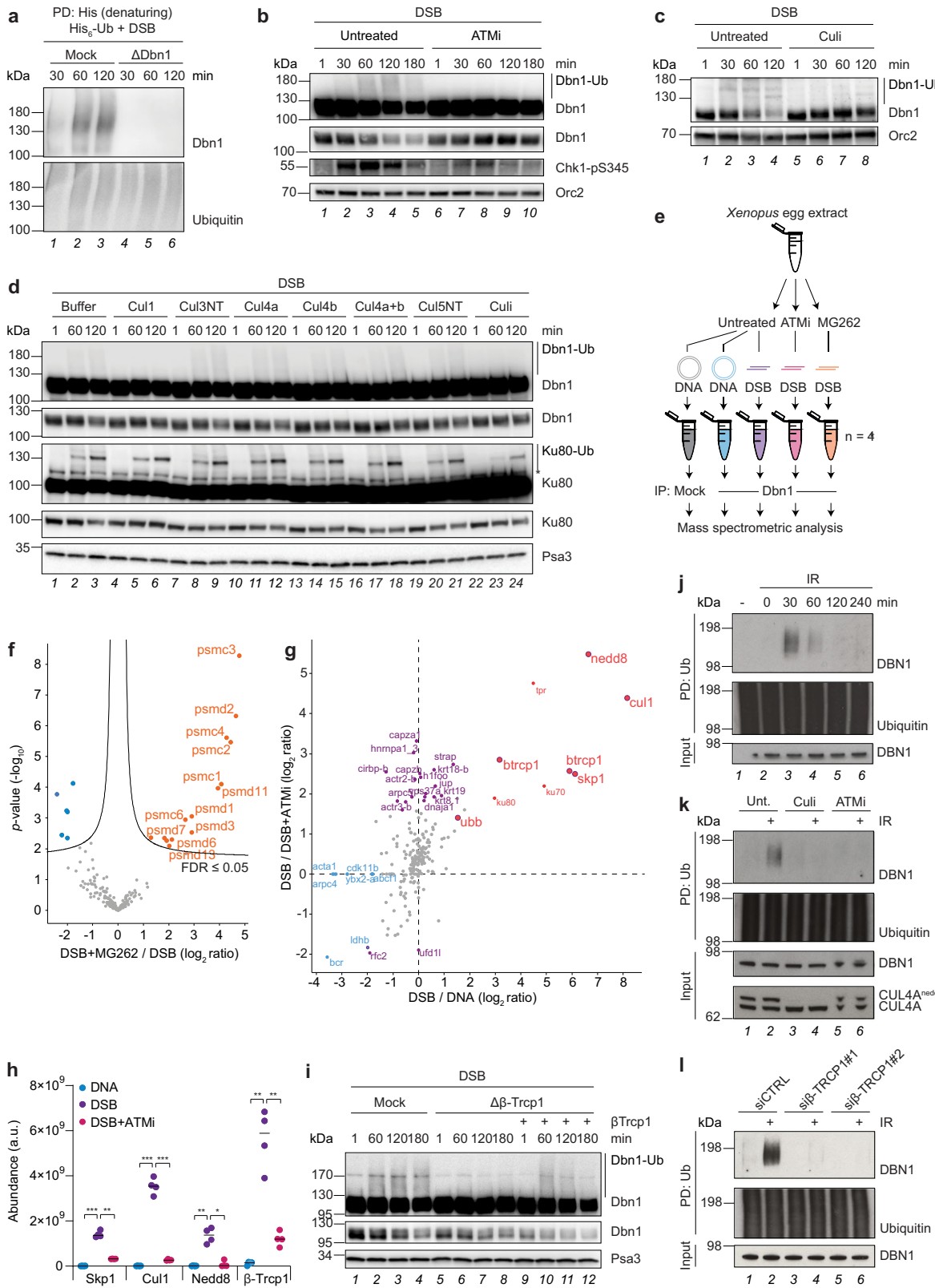

immunoprecipitation-mass spectrometry (IP-MS) experiment in which egg extracts were left untreated or supplemented with either ATM- or proteasome inhibitor before initiating a response by addition of either undamaged or DSB plasmid DNA (Fig. 3e). As shown by the summed peptide abundance across these different conditions, the overall protein content of the samples was similar and enriched over the mock immunoprecipitation control (Supplementary Fig. 3g). Furthermore,

both *Xenopus laevis* isoforms of Dbn1 (Dbn1.S and Dbn1.L) were strongly enriched in the quadruplicate Dbn1-immunoprecipitated samples compared to mock immunoprecipitation (Supplementary Fig. 3h). Consistent with DSB-induced ubiquitylation targeting Dbn1 for proteasomal degradation (Supplementary Fig. 3d), we detected an enriched interaction between Dbn1 and 14 proteasomal subunits in the presence of DSBs and proteasome inhibitor from this unbiased

**Fig. 3 | DSB-induced ubiquitylation of Dbn1 depends on ATM and is mediated by SCF^{β-Trcp1}. a** Western blot analysis of denaturing His-ubiquitin pulldowns from mock- or Dbn1-immunodepleted extracts supplemented with His$_6$-ubiquitin ("His$_6$-Ub") prior to addition of linearized plasmid DNA ("DSB"). Immunodepletion control in Supplementary Fig. 3a. PD, pulldown. **b, c** Western blot analysis of untreated extracts or extracts supplemented with ATM inhibitor ("ATMi", **b**) or neddylation E1 inhibitor ("Culi", **c**) prior to addition of linearized plasmid DNA. **d** Western blot analysis of extracts supplemented with the indicated dominant negative Cullin proteins, neddylation E1 inhibitor or buffer prior to addition of linearized plasmid DNA. *unspecific band. **e** Experimental outline of Dbn1 IP-MS experiment performed in independent reaction quadruplicates. Extracts were untreated or supplemented with ATMi or proteasome inhibitor (MG262) prior to addition of undamaged- ("DNA") or linearized plasmid DNA ("DSB"). Samples were collected for mock- or Dbn1-immunoprecipitation (IP) at 60 min and analysed by MS. **f** Volcano plot analysis comparing proteins enriched from DSB-treated Dbn1 IP-MS samples with *versus* without MG262. Orange and blue dots indicate significantly enriched and -depleted proteins. Significance was determined by two-tailed Student's *t*-test, with permutation-based FDR-control with s0 = 0.1 and 2500 rounds of randomization, to ensure an FDR ≤ 0.05. **g** Scatter plot analysis of the mean abundance difference between proteins enriched with Dbn1-immunoprecipitation from DSB- *versus* undamaged DNA-treated samples plotted against that of DSB-

treated samples without *versus* with ATMi. Red and blue dots indicate proteins significantly enriched and -depleted with Dbn1-immunoprecipitation in the presence of DSBs. Purple dots/outlines indicate proteins significantly changed in enrichment with Dbn1-immunoprecipitation upon ATMi. Significance was determined by two-tailed Student's *t*-test with s0 = 0.1 and permutation-based FDR-control, with 2500 rounds of randomization, to ensure an FDR ≤ 0.05. **h** Abundance of Skp1, Cul1, Nedd8, and β-Trcp1 across the indicated Dbn1 IP-MS conditions. Horizontal lines indicate the median and significance was determined by one-way ANOVA with Tukeys's multiple comparisons test for all conditions shown against DSB-treatment. *p*-values 0.0006, 0.0007, 0.0091, and 0.0062 for DNA *versus* DSB and *p*-values 0.0013, 0.009, 0.0241, and 0.007 for DSB+ATMi *versus* DSB for Skp1, Cul1, Nedd8, and β-Trcp1. a.u., arbitrary units. **i** Western blot analysis of mock- or β-Trcp1-immunodepleted extracts with addition of β-Trcp1 protein or buffer prior to linearized plasmid DNA. **j** Cells exposed to 10 Gy ionizing radiation (IR) were lysed after the indicated times, subjected to ubiquitin pulldown and analysed along with whole cell extracts ("input") by western blot. PD, pulldown; Ub, ubiquitin. **k** Cells were untreated ("Unt.") or treated with Culi or ATMi for 1 h before exposure or not to 10 Gy IR, harvested at 30 min, and processed as described in **j. l** Cells were transfected with control siRNA (siCTRL) or two different β-TRCP1 siRNAs 72 h before exposure or not to 10 Gy IR, harvested at 30 min, and processed as described in **j**. Source data are provided as a Source Data file.

proteomics approach (Fig. 3f and Supplementary Data 4a). Next, we examined the DSB-induced and ATM-dependent Dbn1 interactors, which revealed an enrichment of ubiquitin, Skp1, Cul1, Nedd8 and a single F-box protein, β-Trcp1 (Fig. 3g and Supplementary Data 4a). These proteins were specifically enriched in the presence of DSBs as compared to undamaged DNA, and each was significantly lost upon ATM inhibition (Fig. 3h). Importantly, as the activity of Cullin ubiquitin E3 ligases requires neddylation of the Cullin subunit[54–56], the observed enrichment of Skp1-Cul1-β-Trcp1 along with ubiquitin and Nedd8 (Fig. 3g) supports that Dbn1 interacts with the active SCF^{β-Trcp1} complex upon DSBs.

As our Dbn1 IP-MS experiment corroborated our previous finding that Cul1 is required for DSB-induced ubiquitylation of Dbn1 (Fig. 3c-d), and further suggested that Dbn1 is ubiquitylated by the SCF^{β-Trcp1} complex upon DSBs (Fig. 3g-h), we next investigated the requirement of the SCF substrate recognition factor β-Trcp1 for DSB-induced ubiquitylation of Dbn1. To this end, we raised two antibodies against *Xenopus laevis* β-Trcp1. However, as immunodepletion of β-Trcp1 could not be verified by western blot using these antibodies, we instead confirmed their ability to recognize β-Trcp1 as well as to enrich the Skp1 and Cul1 components of the SCF complex from egg extracts by IP-MS (Supplementary Fig. 3i and Supplementary Data 5). Immunodepletion of β-Trcp1 using either antibody dramatically reduced DSB-induced ubiquitylation of Dbn1 (Supplementary Fig. 3j). The minor co-depletion of Dbn1 with β-Trcp1 immunodepletion suggests an interaction in unperturbed conditions, which is further induced by DSBs (Supplementary Fig. 3j and Fig. 3g-h). We substantiated this by performing denaturing His-ubiquitin pulldowns after addition of recombinant 6xHis-tagged ubiquitin and DSB plasmid DNA to egg extracts, which showed a complete loss of Dbn1 ubiquitylation upon either Cul1- or β-Trcp1 immunodepletion (Supplementary Fig. 3k). Critically, DSB-induced Dbn1 ubiquitylation was restored by addition of recombinant β-Trcp1 protein to β-Trcp1 immunodepleted egg extracts (Fig. 3i and Supplementary Fig. 3l), demonstrating the specific requirement for β-Trcp1 for DSB-induced ubiquitylation of Dbn1. Dbn1 ubiquitylation was also observed in HeLa cells 30 min after DSB formation induced by treatment with ionizing radiation (IR) (Fig. 3j). However, concomitant reduction of unmodified DBN1 was not apparent in the input, suggesting that a minor pool of DBN1 protein is targeted for ubiquitylation upon IR-induced DNA damage in human cells. Nevertheless, the IR induced ubiquitylated DBN1 signal observed was highly reduced upon siRNA-mediated knock-down of DBN1 (Supplementary Fig. 3m), confirming that DNA damage-induced ubiquitylation of DBN1 also occurs in human cells. As observed in egg extracts, DNA

damage-induced DBN1 ubiquitylation was dependent on ATM- and Cullin ubiquitin E3 ligase activity (Fig. 3k). In addition, knock-down of β-Trcp1 by two independent siRNAs also eliminated DBN1 ubiquitylation in response to IR (Fig. 3l). Finally, as DDR signalling and choice of DSB repair mechanism depend on cell cycle stage, we wondered whether DNA damage-induced DBN1 ubiquitylation might also be cell cycle regulated. Ubiquitin pulldowns performed in synchronized cell populations showed that DBN1 was ubiquitylated in response to IR treatment in the asynchronous cell population as well as in G1-phase cells but not in S- or G2-phase cells (Supplementary Fig. 3n). In summary, these data establish a conserved mechanism in which the SCF^{β-Trcp1} complex mediates the DSB-induced and ATM-dependent ubiquitylation of Dbn1 identified by UBIMAX.

## A DDR-specific β-Trcp1 degron drives Dbn1 ubiquitylation

To gain further mechanistic insight to the DDR-dependent and SCF^{β-Trcp1}-mediated ubiquitylation of Dbn1, we sought to identify a putative β-Trcp1 degron in the Dbn1 protein sequence. β-Trcp1 recognizes its substrates via a [D/E/S]-[S/D/E]-G-X-X-[S/E/D] degron motif, in which phosphorylation of both the flanking serine residues is required[57,58]. Indeed, upon scanning the *Xenopus laevis* Dbn1 sequence, we found a putative motif, S-E-G-Y-F-S (amino acids 604-609) located in the unstructured C-terminal region of Dbn1, which is fully conserved across different vertebrate species (Fig. 4a). Intriguingly, the last serine residue in this putative β-Trcp1 degron also forms part of a double ATM consensus [S/T]-Q phosphorylation motif (S609 and S611, respectively)[59,60]. Our Dbn1 IP-MS experiment described above (Fig. 3e) corroborate the assumption that these residues are phosphorylated in a DNA damage and ATM-dependent manner, as we abundantly detected an unmodified Dbn1 peptide containing this putative β-Trcp1 degron in the undamaged condition, while addition of DSB plasmid DNA to egg extracts abrogated detection of this unmodified peptide (Supplementary Fig. 4a, right). Moreover, supplementing egg extracts with an ATM inhibitor prior to the DSB stimulus reenabled the detection of the unmodified peptide. The lack of detection was not due to the general Dbn1 sequence context, as the upstream peptide was detected equally across all conditions (Supplementary Fig. 4a, left). Although the phosphorylated peptide was not detected by MS, we speculate that DSB-induced and ATM-mediated phosphorylation of these SQ motifs could be occurring, and concomitantly enable recognition of Dbn1 by β-Trcp1 specifically in response to DDR activation.

To further investigate the phosphorylation status of the SQ motifs situated in direct connection with the putative β-Trcp1 degron in the

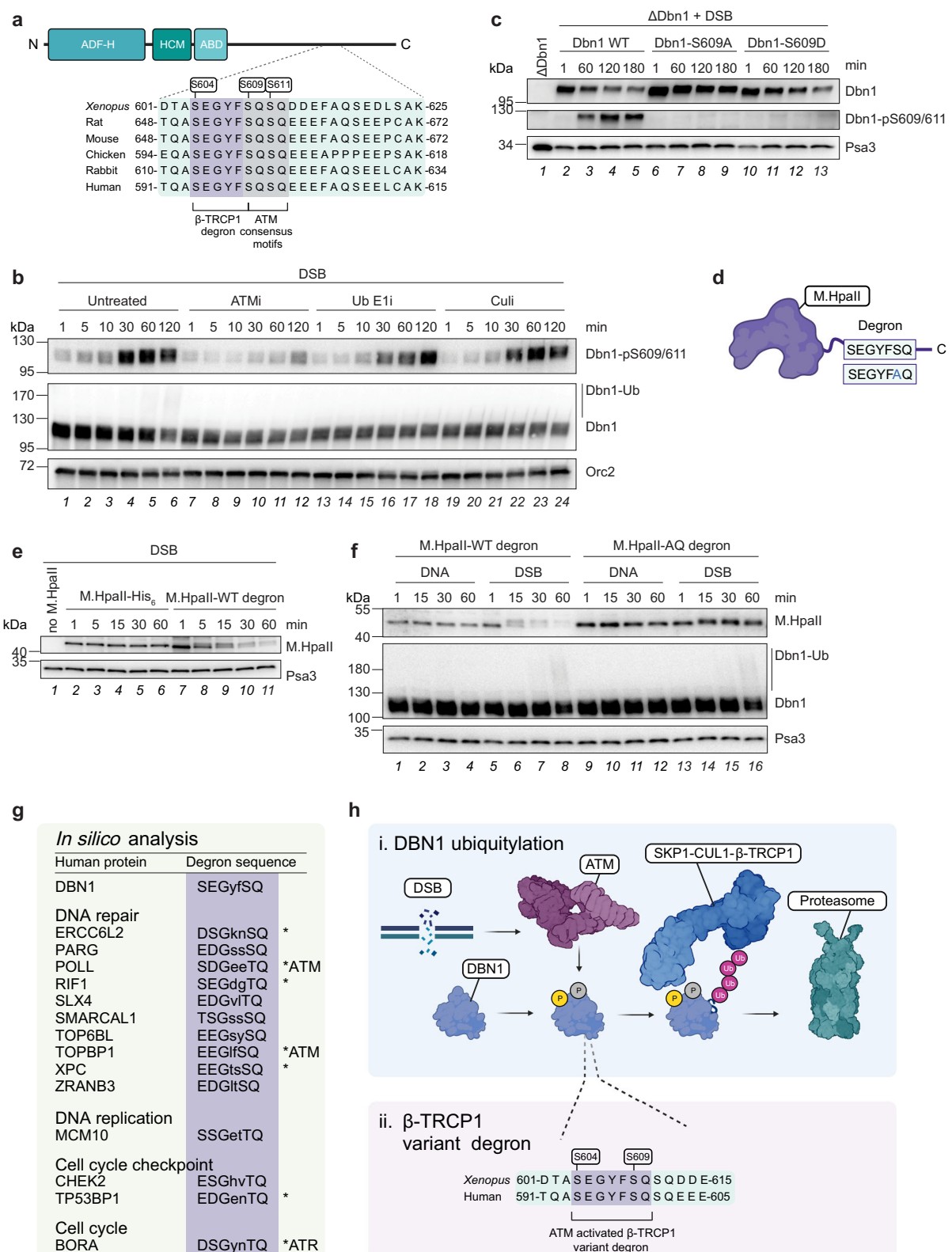

Dbn1 C-terminus, we raised a phospho-specific antibody against these serine residues (Dbn1-pS609/611). Using this antibody, we confirmed phosphorylation of these Dbn1 SQ motifs as early as 5 min following stimulation with DSB plasmid DNA (Fig. 4b, lanes 1-6). While inhibition of either ATM, ubiquitin E1 enzyme, or Cullin E3 ligases prevented ubiquitylation of Dbn, Dbn1-S609/S611 phosphorylation was heavily reduced upon ATM inhibition but remained unaffected by inhibition of

ubiquitin E1 enzyme or Cullin E3 ligases (Fig. 4b). Collectively, this suggests that Dbn1-S609/S611 phosphorylation is mediated by the apical DDR kinase ATM and occurs upstream of Dbn1 ubiquitylation.

To investigate whether phosphorylation of the SQ motif situated in the putative β-Trcp1 degron is required for Dbn1 ubiquitylation, we produced in vitro translated recombinant WT, phosphodeficient (S609A) and phosphomimic (S609D) Dbn1 proteins from rabbit

**Fig. 4 | A variant β-Trcp1 degron is necessary and sufficient for inducing Dbn1 and general protein degradation in response to DSBs. a** Schematic representation of the conserved variant β-Trcp1 degron in the Dbn1 C-terminus. ADF-H, actin depolymerization factor homology; HCM, helical charged motif; ABD, actin-binding domain. **b** Extracts were untreated or supplemented with ATM inhibitor ("ATMi"), ubiquitin E1 inhibitor ("Ub E1i"), or neddylation E1 inhibitor ("Culi"), prior to addition of linearized plasmid DNA ("DSB"). Samples were analysed by western blot at the indicated timepoints. **c** Recombinant Dbn1 WT, S609A, or S609D were added to Dbn1-immunodepleted extracts. Samples were collected from Dbn1-immunodepleted extract prior to addition of recombinant protein, and at the indicated timepoints following addition of protein and linearized plasmid DNA. Samples were analysed by western blot. **d** Schematic representation of the recombinant proteins generated by insertion of WT or mutated variant β-Trcp1 degron at the M.HpaII C-terminus. **e** M.HpaII protein with or without the variant β-Trcp1 degron or buffer ("no M.HpaII") was added to extracts prior to linearized plasmid DNA. Samples were analysed by western blot at the indicated timepoints. **f** M.HpaII protein with WT or AQ-mutated variant β-Trcp1 degron was added to extracts prior to undamaged- ("DNA") or linearized plasmid DNA ("DSB"). Samples were analysed by western blot at the indicated timepoints. **g** In silico analysis of the human proteome revealed numerous proteins containing a potential ATM/ATR-activated β-Trcp1 degron, a subset of which are involved in the DNA damage response. *indicates proteins where the S/TQ site of the putative variant β-Trcp1 degron is known to be phosphorylated. **h, i** DNA damage, such as DSBs, activates the apical DDR kinase ATM, which mediates phosphorylation of the actin-organizing protein Dbn1 at the S609 SQ motif. This primes the connected conserved variant degron (**ii**) for recognition by the F-box protein β-Trcp1, resulting in ubiquitylation by the SCF^β-Trcp1 ubiquitin E3 ligase complex and subsequent proteasomal degradation of the Dbn1 protein. Source data are provided as a Source Data file. Figures **a**, **d**, **g**, and **h** were created with BioRender.com.

reticulocyte lysates. We reasoned that if S609-phosphorylation was required for Dbn1 ubiquitylation, the S609A mutation should render Dbn1 refractory to ATM-mediated phosphorylation and thus preclude recognition by β-Trcp1 for SCF^β-Trcp1-mediated ubiquitylation in response to DSBs. In contrast, the Dbn1-S609D phosphomimic mutant would not require DSB-induced and ATM-dependent phosphorylation for SCF^β-Trcp1-mediated ubiquitylation. To test this hypothesis, we first confirmed that the recombinant WT Dbn1 protein was subjected to ATM-dependent phosphorylation in response to DSBs in egg extracts immunodepleted for endogenous Dbn1 (Supplementary Fig. 4b). However, while recombinant Dbn1 WT and -S609A proteins were readily produced by in vitro translation in rabbit reticulocyte lysates, we were initially not able to produce the Dbn1-S609D mutant. Interestingly, addition of Cullin ubiquitin E3 ligase inhibitor to the in vitro translation reaction enabled production of Dbn1-S609D protein, while addition of proteasome inhibitor resulted in a heavily modified form of Dbn1-S609D (Supplementary Fig. 4c). These observations suggest that recombinant Dbn1-S609D is spontaneously ubiquitylated by a Cullin E3 ligase and subsequently degraded by the proteasome in rabbit reticulocyte lysates, thus hinting at a conserved mechanism for Dbn1 ubiquitylation. To test the relevance of these Dbn1 variants in egg extracts, we supplemented Dbn1 immunodepleted extracts with either the WT, phosphodeficient, or phosphomimic recombinant Dbn1 proteins. Upon DSB addition, WT Dbn1 was phosphorylated at the double SQ motifs, and the amount of unmodified protein correspondingly declined over time (Fig. 4c lanes 2-5). This correlated with WT Dbn1 ubiquitylation upon addition of the DSB plasmid (Supplementary Fig. 4d, lanes 3-4). Importantly, the Dbn1-S609A phosphodeficient mutant was noticeably stabilized in the presence of DSBs (Fig. 4c lanes 6-9) and no damage-induced ubiquitylation was observed (Supplementary Fig. 4d, lanes 7-8). In contrast, the Dbn1-S609D phosphomimic mutant was unstable despite lacking S609/S611 phosphorylation (Fig. 4c lanes 10-13). We also confirmed that this phosphorylation event is conserved and stimulated by DNA damage in HeLa cells transiently transfected with either GFP-tagged WT DBN1 or the corresponding phosphodeficient GFP-DBN1-S599A mutant (Supplementary Fig. 4e).

As recognition of target proteins by β-Trcp1 requires phosphorylation of both serine residues of the target protein degron (Dbn1-S604 and -S609)[57,58], we produced the corresponding single and double phosphodeficient Dbn1 mutants as well as mutants of the second SQ motif immediately downstream of the putative β-Trcp1 degron (S611) (Supplementary Fig. 4f). Mutation of either S604 or S609 rendered recombinant Dbn1 completely stable, despite that the S604 mutation was permissive to DSB-induced S609/S611 phosphorylation (Supplementary Fig. 4f lanes 1-4 and 13-16). In contrast, S611A mutation did not impede Dbn1 degradation, nor did it affect the complete stabilization of Dbn1 conferred by the S609A mutation (Supplementary Fig. 4f lanes 5-12). Together, these data demonstrate

that Dbn1 is targeted for ubiquitylation by the SCF^β-Trcp1 ligase complex through recognition of a variant β-Trcp1 degron, SEGYFSQ, which is specifically sensitive to DNA damage through the direct incorporation of an ATM consensus phosphorylation site.

## The Dbn1 degron is a DDR-sensitive β-Trcp1 variant degron

We wondered whether the DDR-sensitive β-Trcp1 variant degron identified in the Dbn1 C-terminus could function as a general motif for conveying DNA damage-induced ubiquitylation and degradation via the SCF^β-Trcp1 ubiquitin ligase. To investigate this, we cloned the Dbn1 S-E-G-Y-F-S-Q motif onto the C-terminus of the *Haemophilus parainfluenzae* methyltransferase M.HpaII, a protein not native to *Xenopus* egg extracts (Fig. 4d). Strikingly, while the recombinant WT M.HpaII protein was stable in egg extracts challenged by DSB plasmid DNA, the M.HpaII protein tagged with the Dbn1 degron exhibited a mass shift as well as a gradual disappearance of the protein, suggesting the occurrence of DSB-induced phosphorylation and subsequent degradation (Fig. 4e). To confirm degron-targeted degradation of M.HpaII upon DSBs, we additionally cloned the phosphodeficient mutant motif, S-E-G-Y-F-A-Q onto the M.HpaII C-terminus (Fig. 4d) and analysed the stability of the degron-tagged M.HpaII proteins in the presence of either undamaged or DSB plasmid DNA in egg extracts (Fig. 4f). From this, we observed that M.HpaII tagged with the WT Dbn1 degron was destabilized in response to DSB plasmid DNA but remained stable in the presence of undamaged plasmid (Fig. 4f lanes 1-8), confirming that the Dbn1 degron conferred DSB-specific targeting of the M.HpaII protein. Remarkably, M.HpaII tagged with the phosphodeficient Dbn1 degron remained unmodified and stable both in the presence of undamaged and DSB plasmid DNA (Fig. 4f lanes 9-16).

To test whether introduction of the Dbn1 degron onto the C-terminus of M.HpaII confers degradation by the same mechanism as Dbn1, we monitored M.HpaII protein modification and stability upon DSBs in the presence or absence of ATM- or Cullin E3 ligase inhibitors (Supplementary Fig. 4g). Indeed, ATM inhibition abolished phosphorylation of degron-tagged M.HpaII and stabilized the protein in the presence of DSBs, whereas Cullin E3 ligase inhibition was permissive to M.HpaII phosphorylation and stabilized the phosphorylated protein (Supplementary Fig. 4g). Collectively, this confirms that the DDR-sensitive β-Trcp1 degron identified in Dbn1 is transferrable and sufficient for conferring ATM-dependent phosphorylation and subsequent recognition by the SCF^β-Trcp1 ubiquitin ligase complex.

Finally, we wondered if the β-Trcp1 degron identified in Dbn1 could represent a general mechanism for inducing SCF^β-Trcp1-mediated ubiquitylation and degradation of proteins in response to DDR activation. To assess the global distribution of such a variant DDR-β-Trcp1 degron, we carried out an in silico analysis by searching the human proteome for the occurrence of the motif, [D/E/S/T]-[D/E/S]-G-X-X-[S/T]-Q. This analysis identified close to 300 proteins displaying DDR-β-TRCP1 variant degrons, of which we noted several involved in

regulating DNA repair, checkpoint, replication, and cell cycle pathways (Fig. 4g and Supplementary Data 6). Despite that only the identified degrons of BORA, TOPBP1, and POLL are known to be phosphorylated by the apical DDR kinases ATM or ATR[61–64], our in silico analysis shows that 41 proteins have previously been reported phosphorylated at the S/TQ motif of the putative DDR-β-TRCP1 degrons.

Collectively, we find evidence for a wider existence of a DDR-responsive β-TRCP1 variant degron, which could provide a hitherto unappreciated mechanism for inducing a coordinated SCF[β-TRCP1]-mediated ubiquitylation programme in response to DNA damage.

## Discussion

As ubiquitylation is a key signalling modulator involved in regulating most cellular functions, methods for global, unbiased profiling of ubiquitylation events are needed. Here, we presented a method, UBI-MAX, which efficiently and specifically identifies dynamic and quantitative protein ubiquitylation under defined and adaptable conditions of choice in Xenopus egg extracts. We demonstrate that UBIMAX can detect highly DNA-damage specific ubiquitylation events and identify the previously uncharacterized, DSB-induced ubiquitylation of the actin-organizing protein, Dbn1. We unravel the conserved mechanism for this ubiquitylation event and show that it is mediated by the SCF[β-Trcp1] ubiquitin E3 ligase and depends on direct ATM-mediated phosphorylation of a variant β-Trcp1 degron (Fig. 4h). We further show that this variant β-Trcp1 degron is necessary and sufficient for DSB-induced degradation of a model substrate, M.HpaII. Collectively, our work demonstrates UBIMAX's capacity to identify previously unknown and conserved mechanisms of the ubiquitylation response to a defined DNA lesion.

In this study, we have used UBIMAX to investigate ubiquitylation dynamics in response to DNA damage and identify protein ubiquitylation specifically induced by DSBs or DPCs. By detecting proteins previously known to be ubiquitylated upon DNA damage (Fig. 1i) and validating the previously unknown ubiquitylation of the actin-organizing protein Dbn1 (Fig. 2j and Fig. 3a), we show that the denaturing ubiquitin enrichment approach we have utilized for UBIMAX is successful in specifically enriching for ubiquitin-conjugated proteins while eliminating ubiquitin-interacting proteins. This is due to the ease of supplementing Xenopus egg extracts with, in this case, recombinant 6xHis-tagged ubiquitin protein. UBIMAX does not identify a vast amount of ubiquitylated proteins when compared to other proteomic strategies which identify site-level ubiquitylation[13–17], but does identify dynamic protein ubiquitylation in response to specific stimuli while expending 100-fold less starting material[14]. This is due to the advantage of utilizing the very protein-rich Xenopus egg extract system, in which it is possible to generate site-specific DNA lesions of interest and follow the response to these lesions with temporal precision. We have shown that UBIMAX is capable of detecting ubiquitylation events specific to such DNA lesions as well as identifying common DNA damage-related ubiquitylation responses (Fig. 1i, Fig. 2b-c and Supplementary Fig. 2b). Moreover, due to the synchronous nature of Xenopus egg extracts and the ability to easily inhibit or immunodeplete essential proteins, UBIMAX could be utilized to interrogate the ubiquitylation response to defined processes such as DNA replication and mitosis with temporal precision and in the absence of essential protein functions. Furthermore, we found evidence to suggest that UBIMAX can be utilized to investigate other aspects of ubiquitin signalling that have remained technically challenging. For example, we detected DNA damage-induced ubiquitylation of the ubiquitin E3 ligases Chfr and Hltf (Fig. 1i and Fig. 2b-c), which auto-ubiquitylate in response to DNA damage[38,42,43]. In addition to Chfr and Hltf, we observed ubiquitylation of a further 12 ubiquitin E3 ligases in response to DSBs, DPCs or both (Supplementary Data 1 and 3). This indicates the potential of UBIMAX to profile active ubiquitin ligases in response to a condition of choice by detecting ubiquitin E3 ligase auto-ubiquitylation. By further taking

advantage of the possibility to immunodeplete or chemically inactivate specific ubiquitin E3 ligases in egg extract, UBIMAX could be utilized for interrogating ligase-substrate relationships. Finally, we envision that UBIMAX could be employed to provide linkage-specific information about global ubiquitylation as well as be adapted to investigate other ubiquitin-like protein modifications via the direct addition of recombinant 6xHis-tagged linkage-specific ubiquitin mutants or ubiquitin-like proteins to Xenopus egg extracts.

Using UBIMAX, we detected the previously uncharacterized DSB-induced ubiquitylation of the actin-organizing protein Dbn1 (Fig. 1i). Dbn1 binds to and stabilizes actin filaments by preventing depolymerization of actin subunits from the filament barbed end, inhibiting Cofilin-mediated filament severing and inducing actin filament bundling[65–69]. Actin-binding proteins have been shown to associate with DSBs and actin filament polymerization was proposed to regulate DSB localization and repair by homologous recombination (HR) in a manner depending on the DDR[70–74]. However, how the DDR connects to and regulates actin filament dynamics has remained elusive and Dbn1 has not previously been shown to be involved in DSB repair. We show that the apical DDR kinase, ATM, by inducing phosphorylation of Dbn1 in the presence of DSBs, triggers the ubiquitylation and degradation of the Dbn1 protein (Fig. 4b). Furthermore, our Dbn1 IP-MS data indicate that the interaction between Dbn1 and actin filament-related factors (e.g. capza1, capzb) depend on ATM activity while the interaction between Dbn1 and actin (acta1) is reduced in the presence of DSBs (Fig. 3g). A recent study suggests that actin polymerization is required for HR in S/G2 phase of the cell cycle[73] while another reports that actin filaments contacts and are required to position DSBs incurred in G1 awaiting HR repair in S/G2[70]. While we did not detect Dbn1 localization to DSB plasmid DNA in Xenopus extracts (Supplementary Fig. 3f), our data from human cells suggest that DBN1 is mainly ubiquitylated in response to DNA damage occurring in the G1-phase of the cell cycle (Supplementary Fig. 3n). This could indicate a reduced requirement for stabilizing actin filaments and instead allowing for a more dynamic actin filament network at this time of the cell cycle. It would be interesting to understand if DDR-induced degradation of Dbn1 directly impacts nuclear actin filament dynamics and thereby DSB repair.

We show that Dbn1 is ubiquitylated by the SCF[β-Trcp1] ubiquitin E3 ligase through recognition of a variant β-Trcp1 degron (S-E-G-Y-F-S-Q) situated in the unstructured Dbn1 C-terminus (Fig. 3 and Fig. 4c). Recognition requires phosphorylation of both serine residues, of which the one situated in the SQ motif depends on ATM activity (Fig. 4b-c and Supplementary Fig. 4f). We further show that this variant β-Trcp1 degron is necessary and sufficient for degradation of a model substrate, M.HpaII, upon addition of DSBs (Fig. 4e-f). We note, however, that DSB-induced degradation of the model substrate occurred with faster kinetics than Dbn1 (compare Fig. 4b and e). This indicates that additional regulatory mechanisms exist for Dbn1 ubiquitylation. Indeed, the Dbn1 sequence contains additional conserved S/TQ motifs upstream and immediately downstream of the β-Trcp1 degron. Together, these potential ATM phosphorylation sites could form an S/TQ cluster domain, for which it has been suggested that all S/TQ sites need to be phosphorylated in order for the domain to adopt a structure permissive to DNA damage-induced protein-protein interactions[75]. On the other hand, this S/TQ cluster could stimulate phosphatase recruitment and thereby counteract the activation of the degron[76]. In fact, the immediate downstream SQ site (S611) has previously been described as phosphorylated in an ATM-dependent manner in response to oxidative stress in C.elegans (Dbn1-S647), stabilizing the protein, and dephosphorylated by the PTEN phosphatase[77,78]. Finally, Dbn1 is suggested to exist in a closed conformation, which is alleviated by cyclin-dependent kinase-like 5 (Cdk5)-mediated phosphorylation of S142 allowing access to the C-terminus[69]. Indeed, in our Dbn1 IP-MS experiment we detected the phosphorylation of this site in Xenopus

egg extracts (Supplementary Data 4b). Thus, we hypothesize that these additional conserved S/TQ sites surrounding the degron may act as additional regulatory elements to finetune Dbn1 protein stability and actin filament organization.

As the β-Trcp1 degron identified in Dbn1 induced DSB-dependent degradation of a M.HpaII model substrate (Fig. 4e-f), we wondered if this variant β-Trcp1 degron could represent a general mechanism for inducing SCF-mediated ubiquitylation and degradation of proteins in response to DDR activation. We further envision that this variant β-Trcp1 degron could be used to induce specific and timely degradation of essential proteins in response to DSB addition to e.g. investigate specific DNA repair pathways. While a functional β-Trcp1 degron containing an ATR-regulated phosphorylation site has been reported for the mitotic regulator BORA in human cells[61,64], the general occurrence of such a variant degron has not been previously described. Our in silico analysis (Supplementary Data 6) suggests that close to 300 human proteins contain such a variant DDR-β-TRCP1 degron of which 41 have previously been reported to be phosphorylated at the S/TQ motif within the degron. While further studies are required to determine whether these putative degrons are functional, we find evidence for a wider existence of a DDR-responsive β-TRCP1 degron. We envision that such a degron could provide a hitherto unappreciated mechanism for inducing a coordinated SCF$^{\beta\text{-TRCP1}}$-mediated ubiquitylation programme in response to DNA damage and thus regulate DSB repair, DDR-, and checkpoint activity.

## Methods

### Xenopus egg extracts and reactions

All experiments involving animals were approved by the Danish Animal Experiments Inspectorate and are conform to relevant regulatory standards and European guidelines. Egg extracts were prepared using mature (>9 cm) female Xenopus laevis (Nasco Cat #LM0053MX). Preparation of Xenopus high speed supernatant interphase egg extracts (HSS) was performed as described previously[79]. Six to eight female frogs (Nasco) were primed by injection with 80 IU of human chorionic gonadotropin (hCG, Sigma). Two to seven days after priming, frogs were injected with 625 IU of hCG and placed in individual tanks containing 100 mM NaCl. 18-20 h post injection, eggs were collected and used for extract preparation. Eggs were first dejellied in cysteine buffer for 7 min (2.2% cysteine-HCl, pH 7.7), washed three times in 0.5x MMR buffer (50 mM NaCl, 1 mM KCl, 0.25 mM MgSO$_4$, 1.25 mM CaCl$_2$, 2.5 mM HEPES, 0.05 mM EDTA, pH 7.8) and washed three times in ELB sucrose buffer (2.5 mM MgCl$_2$, 50 mM KCl, 10 mM HEPES, 250 mM sucrose, 1 mM DTT, 50 μg/mL cycloheximide, pH 7.8). Eggs were packed for 1 min at 176 × g and crushed for 20 min at 20,000 × g in a swing bucket rotor at 4 °C in the presence of cytochalasin B (2.5 μg/mL), aprotinin (5 μg/mL) and leupeptin (5 μg/mL). Crude interphase extract was recovered post-centrifugation and spun in an ultracentrifuge for 90 min at 260,000 × g at 2 °C following addition of cycloheximide (50 μg/mL), DTT (1 mM), aprotinin (10 μg/mL), leupeptin (10 μg/mL) and cytochalasin B (5 μg/mL). Following centrifugation, the lipid layer on top was removed. The soluble HSS was harvested, snap frozen in 33 μL aliquots and stored at −80 °C.

Reactions were performed at room temperature (RT) using HSS supplemented with 3 μg/mL nocadazole and ATP regeneration mix (20 mM phosphocreatine, 2 mM ATP, 5 μg/mL creatine phosphokinase). Where indicated, HSS was supplemented with various inhibitors and incubated for 10 or 20 min at RT prior to addition of plasmid DNA. To block de novo ubiquitylation, egg extracts were supplemented with 200 μM ubiquitin E1 inhibitor (MLN7243, Active Biochem). Activity of the apical DDR kinases were inhibited using ATM inhibitor (KU-559333, Selleckchem), ATR inhibitor (AZ20, Sigma-aldrich) and DNA-PKcs inhibitor (NU7441, Selleckchem) at final concentrations of 100 μM. Cullin ubiquitin E3 ligase activity was blocked by supplementing egg extracts with 100 μM neddylation E1 enzyme inhibitor (MLN4924, R&D

systems). Proteasome activity was inhibited via addition of 200 μM MG262 (Boston Biochem). Egg extracts were supplemented with recombinant proteins as detailed below and incubated for 10 min at RT before addition of plasmid DNA unless otherwise stated. Where indicated, 6xHis-tagged human recombinant ubiquitin (Boston Biochem) was added to egg extracts at a final concentration of 0.1 μg/μL except to investigate ubiquitin-conjugation linkage type (Supplementary Fig. 3c) and ubiquitylation of Dbn1 WT or -S609A (Supplementary Fig. 4d), for which 6xHis-tagged ubiquitin WT or mutants (Boston Biochem) were added at final concentrations of 1 μg/μL. For testing the Cullin ubiquitin E3 ligase specificity of Dbn1 ubiquitylation, egg extracts were supplemented with recombinant dominant negative Xenopus Cul1, Cul3, Cul4a, Cul4b, and Cul5 proteins at final concentrations of 0.2 μg/μL. Where indicated, in vitro translated Xenopus laevis β-Trcp1, WT and mutant Dbn1 proteins were generally added to egg extracts in a 1:4-10 ratio (see details below). For testing ubiquitylation of a model substrate, recombinant Haemophilus parainfluenzae methyltransferase M.HpaII without or with the WT or AQ mutated β-Trcp1 degron identified in Dbn1 were added to egg extracts in a 1:10 ratio and incubated for 10 or 30 min at RT before addition of plasmid DNA. Reactions were initiated by addition of 15 ng/μL plasmid DNA substrate as indicated. All sampling of Xenopus egg extract reactions were done from individual reactions, except for time course experiments, in which the same reaction was sampled repeatedly. Experimental replicates were sampled from individual reactions.

### Preparation of DNA substrates

The DSB-mimicking plasmid DNA substrate was generated by linearizing pBlueScript II KS (pBS) through enzymatic digestion using XhoI. Circular pBS was used as the undamaged control. To generate a radiolabeled DSB substrate, pBS was first nicked with nb.BsrDI and subsequently radiolabeled with [α-$^{32}$P]dATP via nick translation synthesis by DNA Pol I for 20 min at 16 °C. Radiolabeled pBS was subsequently linearized as described above.

ssDNA-DPC was previously described in[26] as pDPC$^{ssDNA}$. To generate SSB-DPC, we first created pFRT by inserting the specific Flp recognition target site sequence into pBS, by replacing the EcoRI-HindIII fragment with the sequence 5'-AAT TCG ATA AGT TCC TAT TCG GAA GTT CCT ATT CTC TAG AAA GTA TAG GAA CTT CAT CA-3'. For the crosslinking reaction, pFRT was mixed with Flp-nick-His$_6$ in reaction buffer (50 mM Tris-HCl pH 7.5, 50 mM NaCl, 20 μg/mL BSA and 1 mM DTT) and incubated overnight at 30 °C[44].

### Antibodies, Immunodepletion and -detection

Antibodies against Xenopus Mcm6[80] (1:5000), Orc2[81] (1:5000), Rpa[82] (1:1000), as well as M.HpaII[26] (1:1000) were previously described. The antibody against His (631212, lot: 1909019 A, Fisher Scientific, 1:1000) is commercially available. The following antibodies were raised against the indicated peptides derived from Xenopus laevis proteins (New England Peptide, now Biosynth): Dbn1 (Ac-CWDSDPVMEEEEEEEEGGGFGESA-OH, 1:1000), Ku80 (CME-DEGDVDDLLDMM, 1:1000), Cul1 (H2N-MSSNRSQNPHGLKQIGLDQC-amide, 1:2500), Fbxl12 (Ac-CRGIDELKKSLPNSKVTN-OH, 1:2500), Psa3 (Ac-CKYAKESLEEEDDSDDDNM-OH, 1:5000), β-Trcp1-INT (Ac-GQYLFKNKPPDGKTPPNSC-amide, 1:2500), β-Trcp1-N (H2N-MEGFSSSLQPP-TASEREDC-amide, 1:2500), and Dbn1-pS609/611 (Ac-CSEGYF(pS)Q(pS)QDED-amide, 1:2500). These antibodies are available from the authors upon reasonable request. Antibodies against human proteins used in this study include ubiquitin (P4D1, sc-8017), lot: B1422, Santa Cruz), CHK1-pS345 (2341 (133D3), lot: 18, Cell Signalling, 1:1000), DBN1 (TA812128, clone OTI4B1, Thermo Fisher Scientific, 1:1000), CUL4A (2699 S, lot: 1, Cell Signalling, 1:1000), GAPDH (sc-20357 HRP, lot: G2512, Santa Cruz, 1:1000), and CyclinB (610220, lot:84924, BD Biosciences, 1:1000), all of which are commercially available. Secondary antibodies used were Peroxidase AffiniPure Goat Anti-Rabbit IgG (H + L) (111-035-003, lot:

156592, Jackson ImmunoResearch, 1:10.000) or Goat Anti-Rabbit IgG Antibody (H + L) (PI-1000, lot: ZJ0211, Vector Laboratories, 1:10.000) and Peroxidase AffiniPure Rabbit Anti-Mouse IgG (H + L) (315-035-003, lot: 127130, Jackson ImmunoResearch, 1:10.000) or Horse Anti-Mouse IgG Antibody (H + L) (PI-2000, lot: ZJ0428, Vector Laboratories, 1:10.000).

Antibodies raised against *Xenopus* proteins for this study were validated through their ability to immunoprecipitate and consequently immunodeplete the protein in question from *Xenopus* egg extract. β-Trcp1-INT and β-Trcp1-N antibodies were validated by immunoprecipitation from *Xenopus* egg extract followed by mass spectrometry (Supplementary Fig. 3i). All commercially available antibodies were used as per the manufacturer's guidelines and used for the techniques in which they had been validated by the manufacturers.

To immunodeplete *Xenopus* egg extracts, Protein A Sepharose Fast Flow (PAS) (GE Health Care) beads were bound to the indicated antibodies with a stock concentration of 1 mg/mL and at a beads:antibody ratio of 1:4 overnight at 4 °C. IgG antibody was used as the mock control. Beads were washed twice with 500 μL PBS, once with ELB buffer (10 mM HEPES, pH 7.7; 50 mM KCl; 2.5 mM $MgCl_2$; and 250 mM sucrose), twice with ELB buffer supplemented with 0.5 M NaCl, and twice with ELB buffer. One volume of HSS was then depleted by addition of 0.2 volumes of antibody-bound beads and incubating at RT for 15 min with end-over-end rotation, before being harvested. This was repeated one additional round for depletion of Dbn1 and two additional rounds for depletion of Cul1, Fbxl12, and β-Trcp1. Unless otherwise stated, the β-Trcp1-N antibody was used for depletion of β-Trcp1.

For western blot analysis, samples were added to 2x Laemmli sample buffer and resolved on SDS-PAGE gels. Proteins were visualized by incubation with the indicated antibodies and developed using the chemiluminescence function on an Amersham Imager 600 (GE Healthcare) or a Compact 2 developer (Protec). Specifically for western blot analysis of *Xenopus* Dbn1 and human DBN1, the commercially available DBN1 antibody was used in Fig. 2j, Fig. 3c, Fig. 3j–l, Supplementary Fig. 3e, f, Supplementary Fig. 3m, n, and Supplementary Fig. 4e, while the antibody raised against *Xenopus* Dbn1 was used in Fig. 3a, b, Fig. 3d, Fig. 3i, Fig. 4b, c, Fig. 4f, Supplementary Fig. 3a–d, Supplementary Fig. 3j–l, Supplementary Fig. 4b-d, and Supplementary Fig. 4f, g.

## Protein expression and purification

*Xenopus laevis* Dbn1 (.L homolog, Thermo) and β-Trcp1 (.S homolog encoded by bPZ934, a kind gift from Philip Zegerman[83]) was cloned into the pCMV-Sport vector under the Sp6 promoter. The Dbn1 mutant sequences were generated using the KOD Hot Start DNA Polymerase kit (Sigma-Aldrich), according to the manufacturer's instructions. Primers 5′-GTGACAAAAGACACAGCAGCTGAAGGATATTTCAGCCAAT CAC-3′ and 5′-GTGATTGGCTGAAATATCCTTCAGCTGCTGTGTCTTT TGTCAC-3′, 5′-GCAAGTGAAGGATATTTCGCCCAATCACAAGATGAG GACTTTGC-3′ and 5′- GCAAAGTCCTCATCTTGTGATTGGGCGAAATA TCCTTCACTTGC-3′, 5′-GCAAGTGAAGGATATTTCAGCCAAGCACAAGA TGAGGACTTTGC-3′ and 5′- GCAAAGTCCTCATCTTGTGCTTGGCTGAA ATATCCTTCACTTGC-3′, 5′-GCAAGTGAAGGATATTTCGACCAATCACA AGATGAGGACTTTGC-3′ and 5′- GCAAAGTCCTCATCTTGTGA TTGGTCGAAATATCCTTCACTTGC-3′ were used with the Dbn1 WT sequence for generation of the S604A, S609A, S611A, and S609D mutants, respectively. Primers 5′-GCAAGTGAAGGATATTTCGCC CAAGCACAAGATGAGGACTTTGC-3′ and 5′- GCAAAGTCCTCATCTT GTGCTTGGGCGAAATATCCTTCACTTGC-3′, 5′-CAAAAGACACAGCA GCTGAAGGATATTTCGCCCAATCACAAG-3′ and 5′-CTTGTGATTGGG CGAAATATCCTTCAGCTGCTGTGTCTTTTG-3′ were used with the Dbn1 S609A sequence for generation of the S609A/S611A and S604A/ S609A mutants, respectively. The proteins were then expressed by in vitro translation in rabbit reticulocyte lysate. Specifically, two

reactions containing 40 μL TnT SP6 Quick Master Mix (Promega), 2 μL of 1 mM methionine and 1 μg of pCMV-Sport plasmid were incubated for 90 min at RT. For expression of Dbn1-S609D, this reaction was further supplemented with 200 μM neddylation E1 enzyme inhibitor. As a negative control for rescue experiments, a reaction without plasmid DNA was performed. The two reactions were subsequently mixed and concentrated at 4 °C through an Amicon Ultra-0.5 Centrifugal Filter Unit (Millipore) with a 30 kDa cutoff to a total volume of 50 μL. The recombinant proteins used in the experiments presented in Fig. 3i, Fig. 4c, Supplementary Fig. 3l, Supplementary Fig. 4b, Supplementary Fig. 4d, and Supplementary Fig. 4f were produced in this manner and added to egg extract in ratios of 1:4, 1:5, 1:4, 1:10, 1:4, and 1:6.25, respectively.

Plasmids expressing the dominant negative Cullin proteins, Cul1-NT, Cul3a-NT and Cul5a-NT C-terminally tagged with $His_6$- and FLAG tags, were kind gifts from Prof. Alex Bullock and were expressed and purified as previously described[84]. The *Xenopus* gene fragment coding for Cul4b-NT (amino acids 159-510) was synthesised based on Xenbase sequences with addition of $His_6$- and FLAG tag at C-termini, and cloned into pET28 and pET23 vectors, respectively. Cul4a-NT (amino acids 1-396) coding sequence was amplified from *Xenopus* cDNA and cloned into pET23 vector. Both proteins were purified as above.

M.HpaII-$His_6$ was expressed and purified as previously described[45]. To generate the M.HpaII-WT degron and -AQ degron proteins, M.HpaII was first cloned into $pHis_6$-SUMO[85] using primers 5′-TATAGGATCCATGAAAGATGTGTTAGATGATA-3′ and 5′-TATAGAGC TCTCATtcatgccattcaatcttctg-3′, the latter of which contains the sequence for the AviTag. Plasmids encoding M.HpaII-WT degron and -AQ degron were then constructed by addition of the Dbn1 WT or -S609A degron sequences, S-E-G-Y-F-S/A-Q, to the C-terminus of $pHis_6$-SUMO-M.HpaII-Avitag via PCR using primers 5′-atgcGCTAGCGGA TCGGACTCA-3′ and 5′-tataGGTACCTTGGCTGAAATATCCTTCA CTggattggaagtacaggttctcaa-3′ or 5′-tataGGTACCTTGGGCGAAATATC CTTCACTggattggaagtacaggttctcaa-3′ for WT and AQ mutant, respectively. The degron-tagged M.HpaII proteins was subsequently expressed and purified as described in[45] and the N-terminal $His_6$-SUMO-tag cleaved off using the SUMO protease Ulp1. M.HpaII-$His_6$ and M.HpaII-WT degron and -AQ degron proteins were used at final concentrations of 3 ng/μL in egg extract reactions.

## DNA repair assay

For assaying DSB repair in *Xenopus* egg extract, HSS was supplemented with the indicated inhibitors and proteins and reactions initiated by addition of radiolabelled linearized plasmid DNA. At the indicated timepoints, 2 μL reaction was added to 10 volumes of transparent stop buffer (50 mM TrisHCl, pH 7.5, 0.5% SDS, 25 mM EDTA), treated with 1 μL RNase A (Thermo) for 30 min followed by 1 μL Proteinase K (20 mg/mL, Roche) for 1 h at 37 °C. The DNA was purified by phenol/ chloroform extraction, ethanol precipitated in the presence of glycogen (20 mg/mL, Roche), and resuspended in 10 μL of 10 mM Tris buffer (pH 7.5). The DNA was separated by 0.9% native agarose gel electrophoresis and visualized using a phosphorimager. Radioactive signal was quantified using ImageJ (NIH, USA) and quantifications graphed using Prism (GraphPad Software). Within each timepoint, the signal from radioactively labelled repair products (supercoiled, dimers and trimers) were summarised and quantified as mean percent of total radioactive signal within each sample normalized to the input control.

## Denaturing His-ubiquitin pulldown

To enrich ubiquitin-conjugated proteins, Ni-NTA superflow agarose beads (Qiagen) were washed thrice in denaturing pulldown buffer (6 M Guanidine hydrochloride, 0.14 M $NaH_2PO_4$, 4.2 mM $Na_2HPO_4$, 10 mM Tris pH 7.8). At the indicated timepoints, *Xenopus* egg extract reactions supplemented with a final concentration of 0.1 μg/μL $His_6$-ubiquitin, unless otherwise stated, were added to 4 volumes of beads for UBIMAX

and 3.3 volumes for western blot analysis, respectively, in a total of 50 volumes of denaturing pulldown buffer supplemented with 25 mM imidazole and 6.25 mM β-mercaptoethanol and incubated 60 min or overnight at 4 °C with end-over-end rotation. Beads were washed thrice in denaturing pulldown buffer supplemented with 10 mM imidazole and 5 mM β-mercaptoethanol, five times in wash buffer 2 (8 M urea, 78.4 mM NaH$_2$PO$_4$, 21.6 mM Na$_2$HPO$_4$, 10 mM Tris, pH 6.3) and twice in wash buffer 3 (8 M urea, 6.8 mM NaH$_2$PO$_4$, 93.2 mM Na$_2$HPO$_4$, 10 mM Tris, pH 8). For elution of ubiquitin-conjugated proteins for western blot analysis, beads were resuspended in 2x Laemmli sample buffer with 0.5 M EDTA, boiled at 95 °C for 5 min, and eluates separated from the beads by centrifugation through homemade nitex columns.

## UBIMAX

The DSB-UBIMAX experiment included a total of 20 samples: quadruplicate independent reactions across five experimental conditions, including four control conditions for background binding (reactions containing untagged ubiquitin), for de novo ubiquitylation (reactions containing ubiquitin E1 inhibitor) and two reaction controls (containing no DNA or undamaged DNA, respectively) (Fig. 1d). The DPC-UBIMAX experiment included a total of 18 samples: triplicate independent reactions across six conditions, including four control conditions for de novo ubiquitylation and DNA treatment as described above (Fig. 2a).

For MS-based analysis of ubiquitin-conjugated proteins via UBIMAX, ubiquitylated proteins were enriched by denaturing His-ubiquitin pulldown as described above. Following washes, beads were resuspended in 150 μL wash buffer 3 and diluted with two volumes of 10 mM Tris pH 8.5 prior to on-bead digestion of proteins via addition of 500 ng modified sequencing grade Trypsin (Sigma) with incubations of 1 h at 4 °C and then overnight at RT with continuous mixing at 1200 rpm. Eluates containing digested peptides were separated from beads by centrifugation through a 0.45 μM PVDF filter column (Millipore) and cysteines were subsequently reduced and alkylated by addition of 5 mM Tris(2-carboxyethyl)phosphine (TCEP) and 10 mM chloroacetamide (CAA) for 30 min at 30 °C. Tryptic peptides were acidified with 10% trifluoroacetic acid (pH < 4) and purified by C$_{18}$ StageTips prepared in-house[29]. Four plugs of C$_{18}$ material (Sigma-Aldrich, Empore™ SPE Disks, C$_{18}$, 47 mm) were layered per StageTip and activated in 100% methanol, then equilibrated in 80% acetonitrile 10% formic acid, and finally washed twice in 0.1% formic acid. Acidified samples were loaded on the equilibrated StageTips and washed twice with 50 μL 0.1% formic acid. StageTips were eluted into LoBind tubes with 80 μL of 25% acetonitrile in 0.1% formic acid, eluted samples were dried to completion in a SpeedVac at 60 °C, dissolved in 10 μL 0.1% formic acid, and stored at −20 °C until MS analysis.

## Immunoprecipitation for MS analysis

The Dbn1 IP-MS experiment included a total of 40 samples: quadruplicate independent reactions and two technical replicates (as detailed below) across five conditions, including two control conditions for background binding and DNA treatment, respectively (Fig. 3e). The β-Trcp1 IP-MS experiment included a total of 12 samples: triplicate biological replicates across four conditions based on specific immunoprecipitation and including one control (mock IP using an IgG antibody).

For Dbn1 IP-MS analysis, PAS beads were bound to either IgG or the antibody raised against *Xenopus* Dbn1 (stock concentration 1 mg/mL) and at a beads:antibody ratio of 1:2 overnight at 4 °C. Beads were then washed four times with ELB buffer and resuspended in IP buffer (ELB buffer supplemented with 0.35% NP-40, 5 mM NaF, 2 mM sodiumorthovanadate and 5 mM β-glycerophosphate). One volume of the egg extract reactions indicated was then added to 0.4 volumes of antibody-bound beads in a total of 5 volumes of IP buffer and

incubated at 4 °C for 1 h with end-over-end rotation. Beads were washed thrice in ELB buffer supplemented with 0.25% NP-40 and 500 mM NaCl, transferred to LoBind tubes, and washed a further three times with ELB buffer supplemented with 500 mM NaCl. Immunoprecipitation of Cul1 and β-Trcp1 was performed with the indicated antibodies in essentially the same manner except, that one volume of unstimulated HSS was added to 0.67 volumes of antibody-bound beads in 3.67 volumes of ELB buffer and incubated 3 h at 4 °C, then washed thrice in ELB buffer supplemented with 0.25% NP-40, followed by three washes in ELB buffer. Beads were resuspended in 50 mM ammonium bicarbonate and samples subjected to on-bead digestion via addition of 100 ng modified sequencing grade Trypsin (Sigma) with incubations of 1 h at 4 °C and then overnight at 37 °C with continuous mixing at 1200 rpm. Eluates were separated from beads by centrifugation through a 0.45 μM PVDF filter column (Millipore) and subjected to further 1 h of in-solution digestion by addition of additional 100 ng Trypsin. Cysteines were subsequently reduced and alkylated by addition of 5 mM TCEP and 10 mM CAA for 30 min at 30 °C and tryptic peptides were acidified and desalted on C$_{18}$ StageTips as described above. Additionally, for the Dbn1 IP-MS experiment, samples were divided into two of which one half was desalted using low-pH clean-up as described above, while the other half was desalted using high-pH clean-up[86]. High-pH clean-up was done essentially as described above except StageTips were equilibrated using 100 μL of methanol, 100 μL of 80% acetonitrile in 200 mM ammonium hydroxide, and two times 75 μL 50 mM ammonium. Samples were supplemented with 0.1 volumes of 200 mM ammonium hydroxide (pH > 10), just prior to loading them on StageTips. The StageTips were subsequently washed twice with 150 μL 50 mM ammonium hydroxide, and afterwards eluted using 80 μL of 25% acetonitrile in 50 mM ammonium hydroxide.

## Whole proteome analysis

Volumes corresponding to 100 μg protein from three different batches of HSS extracts were diluted 100-fold in denaturing digestion buffer (6 M Guanidine hydrochloride, 100 mM Tris, 5 mM TCEP, 10 mM CAA, pH 8.5), sonicated and digested using Lys-C (1:100 w/w; Wako) for 3 h at RT. Digestions were subsequently diluted with two volumes of 25 mM Tris pH 8.5 and further digested by addition of modified sequencing grade Trypsin (1:100 w/w) overnight at 37 °C. Tryptic peptides were fractionated on-StageTip at high-pH essentially as described previously[86]. Briefly, StageTips were conditioned with 100 μL methanol, equilibrated with 100 μL 80% acetonitrile in 200 mM ammonium hydroxide, and washed twice with 75 μL 50 mM ammonium hydroxide. The pH of the digested samples was raised by addition of ammonium hydroxide to a final concentration of 20 mM. Samples were loaded on StageTips, and washed twice with 75 μL 50 mM ammonium hydroxide. Peptides were eluted from StageTips as eight fractions (F1-8) using 80 μL of 2, 4, 7, 10, 13, 17, 20, and 40% acetonitrile in 50 mM ammonium. All fractions were dried to completion in LoBind tubes, using a SpeedVac for 3 h at 60 °C, after which the dried peptides were dissolved in 12 μL of 0.1% formic acid. The total number of samples analysed was 24 (eight fractions of three independent biological replicates of HSS).

## MS data acquisition

MS samples were analysed on an EASY-nLC 1200 system (Thermo) coupled to either a Q Exactive HF-X Hybrid Quadrupole-Orbitrap mass spectrometer (Thermo) for the total proteome and DSB-UBIMAX samples, or an Orbitrap Exploris 480 mass spectrometer (Thermo) for the remaining MS experiments in this study. Separation of peptides was performed using 15 cm columns (75 mm internal diameter) packed in-house with ReproSil-Pur 120 C$_{18}$-AQ 1.9 mm beads (Dr. Maisch). Elution of peptides from the column was achieved using a gradient ranging from buffer A (0.1% formic acid) to buffer B (80% acetonitrile

in 0.1% formic acid), at a flow rate of 250 nL/min. For total proteome and DSB-UBIMAX samples, gradient length was 77 min per sample, including ramp-up and wash-out, with an analytical gradient of 55 min ranging from 5% to 25% buffer B for the total proteome and 52.5 min ranging from 10% to 25% buffer B for DSB-UBIMAX. For the remaining MS experiments, gradient length was 80 min per sample, including ramp-up and wash-out, with an analytical gradient of 57.5 min ranging in buffer B from 10% to 40% for DPC-UBIMAX samples and 52.5 min ranging in buffer B from 10% to 30% for IP-MS samples. The columns were heated to 40 °C using a column oven, and ionization was achieved using either a NanoSpray Flex ion source (Thermo) for the total proteome and DSB-UBIMAX, or a NanoSpray Flex NG ion source (Thermo) for the remaining MS experiments. Spray voltage was set at 2 kV, ion transfer tube temperature to 275 °C, and RF funnel level to 40%. Samples were measured using 5 μL injections with different technical settings as detailed in the following. Measurements were performed with a full scan range of 300-1,750 m/z, MS1 resolution of 60,000, MS1 AGC target of 3,000,000, and MS1 maximum injection time of 60 ms for the total proteome and DSB-UBIMAX samples and MS1 resolution of 120,000, MS1 AGC target of "200" (2,000,000 charges), and MS1 maximum injection time "Auto" for the remaining MS experiments. Precursors with charges 2-6 were selected for fragmentation using an isolation width of 1.3 m/z and fragmented using higher-energy collision disassociation (HCD) with a normalized collision energy of 28 for total proteome and DSB-UBIMAX and 25 for the remaining MS experiments. Precursors were excluded from resequencing by setting a dynamic exclusion of 45 s for total proteome and DSB-UBIMAX samples and 60 s with an exclusion mass tolerance of 20 ppm, exclusion of isotopes, and exclusion of alternate charge states for the same precursor for the remaining MS experiments. MS2 AGC target was set to 200,000 and minimum MS2 AGC target to 20,000 for total proteome and DSB-UBIMAX samples and MS2 AGC target to "200" (200,000 charges) with an MS2 intensity threshold of 230,000 or 360,000 for DPC-UBIMAX, Dbn1 IP-MS and β-Trcp1 IP-MS, respectively. For the total proteome samples, MS2 maximum injection time was 55 ms, MS2 resolution was 30,000, and loop count was 12. For DSB-UBIMAX samples, MS2 maximum injection time was 90 ms, MS2 resolution was 45,000, and loop count was 9. The MS2 settings were similar for the DPC-UBIMAX samples, except MS2 maximum injection time was set to "Auto". This was also the case for the IP-MS samples, but while the Dbn1 IP-MS samples were aquired with MS2 resolution of 45,000 and a loop count of 9, the β-Trcp1 IP-MS samples were aquired with MS2 resolution of 15,000 and a loop count of 18. For the DPC-UBIMAX and IP-MS experiments, Monoisotopic Precursor Selection (MIPS) was enabled in "Peptide" mode.

## MS data analysis

All MS RAW data were analysed using the freely available MaxQuant software[87], version 1.6.0.1. Default MaxQuant settings were used, with exceptions specified below. For generation of theoretical spectral libraries, the *Xenopus laevis* FASTA database was downloaded from Uniprot on the 13th of May 2020 for the total proteome and UBIMAX experiments and on the 3rd of September 2021 for the IP-MS experiments. In silico digestion of proteins to generate theoretical peptides was performed with trypsin, allowing up to 3 missed cleavages. Allowed variable modifications were oxidation of methionine (default), protein N-terminal acetylation (default) for all samples. For UBIMAX experiments, ubiquitylation of lysine and cysteine as well as carbamidomethyl on cysteine were additionally included as variable modifications. For IP-MS experiments, ubiquitylation of lysine and phosphorylation of serine and threonine were additionally allowed. Maximum variable modifications per peptide was reduced to 3. LFQ[30] and iBAQ was enabled. For DSB-UBIMAX, LFQ was applied separately within parameter groups defined by sample type (controls *versus* ubiquitin target enriched samples). Stringent MaxQuant 1% FDR data

filtering at the PSM and protein-levels was applied (default). Second peptide search was enabled. Matching between runs was enabled, with an alignment window of 20 min and a match time window of 1 min. For total proteome analysis, matching was only allowed within the same fractions and for IP-MS experiments within replicates of the same sample group. For the Dbn1 IP-MS experiment, dependent peptide search was additionally enabled.

## MS data annotation and quantification

The *Xenopus laevis* FASTA database downloaded from UniProt lacked comprehensive gene name annotation. Missing or uninformative gene names were, when possible, semi-automatically curated, as described previously[34]. Briefly, informative gene names were drawn from Uniprot, otherwise Xenbase, otherwise the Session et al. database[88], otherwise RefSeq (via Xenbase), and otherwise InterPro annotations were used. Quantification of the MaxQuant output files ("proteinGroups.txt") was performed using Perseus software[89], as was Pearson correlation (calculated using linear regression), coefficients of variation, Principal Component, and heirarchical clustering analyses. Enrichment analysis, also performed using Perseus software, was based on terms from the Gene Ontology (GO) and Keywords databases as annotated for the ubiquitylated proteins present in the "DSB-induced" cluster of the heirarchical clustering analysis (Fig. 1h) and compared to those of a total *Xenopus laevis* proteome derived from UniProt with isoforms excluded based on annotated gene names. For protein network analysis, the STRING *Xenopus* (Silurana) *tropicalis* database[90] was queried at a confidence level of 0.7. For quantification purposes, all protein LFQ intensity values were log2 transformed, and filtered for presence in four out of four replicates in at least one experimental condition for the DSB-UBIMAX experiment; three out of three for total proteome, DPC-UBIMAX and β-Trcp1 IP-MS experiments and four out of eight in the Dbn1 IP-MS experiment. Missing values were imputed below the global experimental detection limit at a downshift of 1.8 and a randomized width of 0.3 (in log2 space; Perseus default). For the data presented in volcano- and scatter plots, statistical significance of differences was tested using two-tailed Student's *t*-testing, with permutation-based FDR control applied at s0 values of 0.1 and proteins were filtered to be significantly enriched or depleted at FDR < 5%. Only proteins testing significantly enriched over both "no His" and "Ub E1i" controls for UBIMAX and over the mock control for the IP-MS experiments, respectively, with FDR < 5% when tested using one-tailed Student's *t*-testing, with permutation-based FDR control applied at an s0 value of 0.1, were considered for further analysis. All tested differences, *p*-values, and FDR-adjusted *q*-values are reported in Supplementary Data 1, 3, 4 and 5. For hierarchical clustering analysis of ubiquitylated proteins changing with DNA treatment only robustly changing proteins were considered for the analysis. Robustly changing was defined as proteins increased compared to the median in all four replicates of at least one sample group and decreased compared to the median in all four replicates of another sample group. Quantification of individual peptides or summed peptide abundances were derived from the MaxQuant output files ("evidence.txt")

The original analysis of the total proteome included triplicate samples of both HSS and nucleoplasmic extract (NPE). However, as all *Xenopus* egg extract experiments in this study are performed in HSS, only this extract was included in the further analysis of the total proteome. The original analysis of the DPC-UBIMAX experiment included four replicates but one replicate was excluded due to significant technical variance. The samples of the Dbn1 IP-MS experiment were aquired as two technical replicates on the basis of C18 StageTip method (see above), with runs resulting from high-pH StageTip clean-up denoted by "H" in the raw files and replicates 01-04 in the analysis, while runs resulting from low-pH StageTip clean-up is denoted by "L" in the raw files and replicates 05-08 in the analysis. Furthermore, this experiment originally included a condition treated with ubiquitin E1

enzyme inhibitor and DSB-mimickcking plasmid DNA, but as this condition did not yield significant additional information, it was excluded for further analysis. Finally, in Fig. 3h, only the samples resulting from high-pH StageTip clean up are presented.

## Plasmid pulldown

Plasmid pulldown assays were performed as previously described[26]. 9 µL/pulldown of streptavidin-coupled beads (Dynabeads M-280, Invitrogen) were washed twice with wash buffer 1 (50 mM Tris pH 7.5, 150 mM NaCl, 1 mM EDTA pH 8, 0.02% Tween-20). Biotinylated LacI was added to the beads at 18 pmol/9 µL of beads, and incubated at RT for 1 h. The beads were washed four times with pulldown buffer (10 mM HEPES pH 7.7, 50 mM KCl, 2.5 mM MgCl$_2$, 250 mM sucrose, 0.02% Tween-20) and resuspended in 80 µL of the same buffer and stored on ice. At the indicated timepoints, 20 µL of reaction was withdrawn and gently mixed with the beads. The suspension was rotated for 30 min at 4 °C. The beads were then washed twice with wash buffer 2 (10 mM HEPES pH 7.7, 50 mM KCl, 2.5 mM MgCl$_2$, 250 mM sucrose). After washing, beads were resuspended in 20 µL of 2x Laemmli sample buffer and equal volume of samples were resolved by SDS-PAGE for western blot analysis.

## Cell culture and ubiquitin pulldown

Human HeLa cell lines (CCL-2, ATCC) were cultured under standard conditions at 37 °C and 5% CO$_2$ in DMEM (Thermo) containing 10% FBS (v/v) and penicillin-streptomycin (Thermo). Cells were regularly tested negative for mycoplasma infection. To block Cullin ubiquitin E3 ligase activity, 1 µM neddylation E1 enzyme inhibitor (MLN4924, R&D systems) was added to the cell culture medium. To prevent ATM activity, the cell culture medium was supplemented with 10 µM ATM inhibitor (KU-55933, Selleckchem). Where indicated, cells were subjected to 10 Gy of ionizing radiation using a Smart X-ray machine (Yxlon). For synchronization of HeLa cell cultures, cells were treated with Nocodazole for 16 h before being released by mitotic shake-off for 4 h to obtain the G1 cell fraction. A double 16 h thymidine block was performed, before a 4 h release to obtain the S-phase cell fraction and an 8 h release for the G2-phase fraction. All sampling of human cells were derived from individual cell cultures. No repeated measurements of individual cell cultures were performed.

For western blot analysis, cells were lysed in RIPA buffer (140 mM NaCl, 10 mM Tris-HCl pH 8.0, 0.1% sodium deoxycholate (w/v), 1% Triton X-100 (v/v), 0.1% SDS (w/v), 1 mM EDTA, 0.5 mM EGTA). For ubiquitin pulldown, cell lysates were prepared in lysis buffer (50 mM Tris pH 8.0, 1 M NaCl, 5 mM EDTA, 1% IGEPAL, 0.1% SDS). Lysates were sonicated once for 20 s with an amplitude of 75% on a hand held sonicator, before spinning down at full speed for 20 min in a 4 °C cooled centrifuge. Ubiquitin enrichment was performed using Halo-tagged MultiDsk TUBE[91] preincubated with HaloLinkTM resin (G1913, Promega) for 1 h with rotation at RT in binding buffer (100 mM Tris pH 7.5, 150 mM NaCl, 0.05% IGEPAL). Excess protein was washed off with binding buffer supplemented with 1 mg/mL BSA. Cell lysates were added to the MultiDsk-bound resin and incubated with rotation overnight at 4 °C. Samples were washed four times with MultiDsk lysis buffer before being eluted in 2x Laemmli sample buffer by boiling for 5 min at 95 °C.

All lysis and wash buffers were supplemented with 1 mM dithiothreitol (Sigma), complete EDTA-free protease inhibitor Cocktail Tablets (Roche), 1.25 mM N-ethylmaleimide (Sigma) and 50 µM PR-619 (Calbiochem).

## siRNA and plasmid transfections

The following siRNAs were used in this study to knock-down the expression of selected proteins: Non-targeting control (siCTRL) 5′- GGGAUACCUAGACGUUCUA-3′, siDBN1 5′-GGAGCUUUCGGG ACACUUtt −3′, siβ-Trcp1#1 5′-GUGGAAUUUGUGGAACAUCtt-3′,

siβ-Trcp1#2 5′-AAGUGGAAUUUGUGGAACAUCtt-3′. All siRNAs were used at 20 nM concentrations and transfected with Lipofectamine RNAiMAX reagent (Thermo) according to the manufacturer's instructions.

For transient overexpression of WT or S599A mutated DBN1, Full-length DBN1 (human) cDNA was inserted into pcDNA4/TO-EGFP through Gateway® cloning. The DBN1-S599A phospho-mutant was generated by Q5 mutagenesis with sgRNA-DBN1 5′-CCAGTGAGGGGTACTTCGCTC AATCACAGGAGG-3′. Plasmids were transfected with Lipofectamine 2000 (Thermo) according to the manufacturer's instructions.

## In silico analysis of variant β-TRCP1 degron

A list containing the identity, motif sequence and sequence position of the human proteins containing a variant β-TRCP1 degron was generated by submitting the motif [DEST]-[DES]-G-x(2)-[ST]-Q to the Scan-Prosite tool[92] to scan against the UniProt Homo Sapiens (taxonomy ID: 9606) database. To this list was mapped the phosphorylation status and kinase relationship of the [ST]-Q site, if known, as retrived from the Phospho.ELM[93] and PhosphoSitePlus v6.7.1.1[94] databases.

## Statistics and reproducibility

Bioinformatic analysis of mass spectrometry data was carried out with the Perseus software. Statistical significance of differences was tested using Student's $t$-testing, with permutation-based FDR-control applied at an s0 value of 0.1 with 2500 rnadomizations. Auto-radiographs were quantified using ImageJ. Graphs and the statistical tests displayed in them were done in Prism (GraphPad) using the statistical tests indicated for each analysis. For all statistical analyses: *$p$-value ≤ 0.05, **$p$-value ≤ 0.01; ***$p$-value ≤ 0.001; ****$p$-value ≤ 0.0001. Error bars represent the standard error unless otherwise stated. For all figures in which a representative experiment is shown, experiments were independently repeated at least twice with similar results.

## Reporting summary

Further information on research design is available in the Nature Portfolio Reporting Summary linked to this article.

## Data availability

For generation of theoretical spectral libraries to use with analysis of MS raw data, the *Xenopus laevis* FASTA database was downloaded from Uniprot on the 13th of May 2020 for the total proteome and UBIMAX experiments and on the 3rd of September 2021 for the IP-MS experiments. The mass spectrometry proteomics data generated in this study have been deposited in the ProteomeXchange Consortium via the PRIDE partner repository[95] under the accession code PXD042086 (UBIMAX, IP-MS and total proteome experiments). Source data are provided with this paper.

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

## Acknowledgements

We thank members of the Nielsen and Duxin laboratories for feedback on the manuscript. We thank Philip Zegerman for the β-Trcp1 encoding plasmid and Alex Bullock for the plasmids encoding Cul1-NT, Cul3a-NT

and Cul5a-NT. The Novo Nordisk Foundation Center for Protein Research is supported financially by the Novo Nordisk Foundation (grant agreement NNF14CC0001). The work carried out in this study was in part supported by the Novo Nordisk Foundation Center for Protein Research, the Novo Nordisk Foundation (NNF14CC0001, NNF13OC0006477; M.L.N., and NNF21OC0071976; J.P.D.), The Danish Council of Independent Research (8020-00220B, M.L.N.), The Danish Cancer Society (R146-A9159-16-S2, M.L.N.). The proteomics technology applied was part of a project that has received funding from the European Union's Horizon 2020 research and innovation programme under grant agreement EPIC-XS-823839 (M.L.N.). J.A.G. was funded by the European Union's Horizon 2020 research and innovation programme (Marie-Skłodowska-Curie grant agreement no. 860517 (UBIMOTIF)). This project has also received funding from the Medical Research Council (MR/K007106/1, A.G.). Figures 4a, 4d and 4g-h were created using BioRender.com.

## Author contributions

C.S.C., J.P.D., M.L.N. conceived the project. C.S.C., I.A.H., M.L.N. designed the MS experiments and UBIMAX method. C.S.C. performed the MS experiments. C.S.C., I.A.H. analysed the MS data. C.S.C., I.A.H., J.P.D., M.L.N. interpreted the MS data. C.S.C., E.S.K., J.P.D. designed the *Xenopus* egg extract experiments. C.A., A.G. produced the recombinant dominant negative Cullin proteins. Z.F. generated the SSB-DPC plasmid DNA substrate. C.S.C., E.S.K. performed the *Xenopus* egg extract experiments. C.S.C., E.S.K., J.P.D., M.L.N. interpreted the *Xenopus* egg extract experiments. C.S.C., E.S.K., J.A.G., N.M., J.P.D. designed the human cell experiments. J.A.G. performed the human cell experiments. C.S.C., E.S.K., J.A.G., N.M., J.P.D., M.L.N. interpreted the human cell experiments. C.S.C. performed the in silico analysis of the variant β-TRCP1 degron. C.S.C. prepared the figures and wrote the manuscript with feedback from J.P.D. and M.L.N. E.S.K. created the models of Figs. 4a, 4d and 4g-h. All authors provided critical review of the manuscript.

## Competing interests

The authors declare no competing interests.
