## [Peer Review File · Nature Communications]

Profiling ubiquitin signaling with UBIMAX reveals DNA damage- and SCF β -TRCP1-dependent ubiquitylation of the actin-organizing protein Dbn1Reviewer #1 (Remarks to the Author):

This paper by Colding-Christensen et al. describes a new technique (UBIMAX) to identify proteins whose ubiquitylation is induced by DNA damage in *Xenopus* oocyte extracts. Extracts supplemented with recombinant His-Ub are incubated with a defined DNA lesion, and ubiquitylated proteins are enriched by denaturing pulldown followed by mass spectrometry. Using this technique, the authors find that the actin binding protein Dbn1 is ubiquitylated by Cul1- β Trcp in response to double strand breaks (DSBs) at an ATM-stimulated degron to trigger proteasomal degradation of Dbn1.

The paper does a nice job characterizing DSB-induced ubiquitylation of Dbn1 by Cul1- β TRCP and showing the ATM-regulated degron is sufficient for inducing ubiquitylation of a heterologous protein. The mass spectrometry data is nicely controlled (with "no His" and E1 inhibitor conditions). The DSB experiments are performed in quadruplicate with good correlation, and the data are analyzed well. However, like prior tagged Ub approaches, the technique does not identify many ubiquitylated proteins (see point 1), does not identify sites of ubiquitylation, and enriches nonspecific proteins. Also, there is no functional characterization of Dbn1 ubiquitylation in DNA repair.

1. The authors identify a total of only 786 ubiquitylated proteins from their quadruplicate DSB experiments (text page 6, Fig S1G). With an FDR <0.05, they find only 23 proteins induced by DSBs (Figs 1H, S1G, S1I; Table S1). With a more stringent FDR <0.01, they find 7 proteins (Fig S2B, Table S1). Similarly, ssDNA-DPC (Figs 2B, S2B) induces only 19 proteins (at FDR < 0.05) and 8 proteins (at FDR < 0.01), while SSB-DPC (Figs 2C, S2B) induces 19 proteins (at FDR < 0.05) and 8 proteins (at FDR < 0.01). Newer techniques like diGly peptide enrichment or UbiSite (Akimov NSMB 2018) were developed to identify ubiquitylated proteins more efficiently than tagged ubiquitin/UBL approaches (Meierhofer J Prot Res 2008; Danielsen Mol Cell Prot 2011; Ma et al. Hunter Mol Cell Prot 2014). This is a drawback of their approach.

2. In Fig S1F, enriched peptide abundance for the E1 inhibitor and "no His" controls (bars 4 and 5) is comparable to that for the His-Ub enrichment conditions (bars 1-3). This suggests the majority of peptides purified by Ni-NTA beads are binding non-specifically, which highlights a limitation of their approach.

3. In Fig 1F, E1 inhibition reduces the percentage of ubiquitin peptides enriched on nickel beads, which again suggests a high prevalence of non-specific binding to beads. If the majority of peptides are enriched via specific pulldown by covalent attachment to His-Ub, E1 inhibition should increase the ratio of ubiquitin peptides to non-ubiquitin peptides (by decreasing covalent attachment of His-Ub to other non-ubiquitin proteins), but this does not occur.

4. They should perform UBIMAX in extracts treated with a proteasome inhibitor. This will increase sensitivity for ubiquitylation events that cause degradation by the proteasome

5. The connection of DNA damage to Dbn1 degradation is interesting given recent results showing that actin regulates DSB localization, resection, and repair (Belin eLife 2015, Schrank Nature 2018, Caridi Nature 2018). The authors should test whether Dbn1 depletion (and importantly addback of S609A Dbn1) affects DSB repair. They should check their NHEJ assay (Fig S1B-C) where they have shown E1 inhibition inhibits repair. If no phenotype is found there, they should look for other evidence of defects in DSB repair. They could check DSB resection (Liao NAR 2012) or resection/recombination-based repair assays in *Xenopus* (see references cited in Graham, Meth Enzym 2017). Given that the Dbn1 ubiquitylation event is conserved in HeLa cells, they could also check HR, resection, RAD51 foci formation, or other repair assays in mammalian cells.

6. Along these lines, it has been shown that actin and actin regulators localize to chromosomal DSBs in *Xenopus* extracts (Schrank Nature 2018). They should test whether Dbn1 is recruited to chromosomal DSBs (like Schrank et al) or to their DSB plasmid (as they do in Larsen, Mol Cell 2019)

7. Fig 3I: they need to blot for Cul1 and Skp1 to determine the extent to which β Trcp

immunodepletion reduces the levels of Cul1 and Skp1. To show specificity, they add back β Trcp, but β Trcp is translated in reticulocyte lysates which may contain Cul1 and Skp1.

8. The authors should explain why they observe conflicting results for Dbn1 and Ku80 degradation in different figures. This must be reconciled.

a. DSB induces Ku80 degradation at 120 min in Fig S3E but not in Fig 3D.

b. They see DSB-induced WT Dbn1 degradation in Fig 4C but not in Fig S4B. In both cases of these cases, Dpn1b was depleted and then WT Dbn1 was added back

9. Fig 3G, Table S4: why is ATMi decreasing the majority of Dbn1 interactions (with proteins like actin and the actin regulators capza1, capzb, arpc3a, aprc5, arpc2b). It makes sense that ATMi would block interaction with β Trcp, Skp1, and Cul1, but I would expect ATMi to increase basal interaction with actin (and its binding partners) because ATMi should stabilize Dbn1. Also, their MS data in Table S4 (corresponding to Fig 3G) show that Dbn1 levels are not higher upon ATM inhibition. These results suggest the ATMi IP sample may have overall lower protein/extract input, and the authors should address this.

10. Fig S3H, Fig 3I: why are basal levels of Dbn1 (at 1 min) lower in β Trcp-depleted extracts? It is nice how degradation is inhibited (i.e., Dbn1 levels are stable over time), but why do Dbn1 levels start out lower? This becomes an issue in Figs S3I/J, where it looks like β Trcp depletion is abolishing Dbn1 ubiquitylation but not increasing Dbn1 levels. This may result from loss of Dbn1 from extracts by immunodepletion of β Trcp. If so, this would suggest β Trcp binds Dbn1 in the absence of DNA damage. The authors should address this observation and implications in the text.

11. Fig 3J, K, L: why is IR not inducing Dbn1 degradation in HeLa cells? If the authors believe Dbn1 ubiquitylation serves a different purpose in HeLa, they should acknowledge and address this in the manuscript.

12. Fig S4A, text on p.12: "lack of detection was not due to the general Dbn1 sequence context, as the upstream peptide was detected equally across all conditions." I do not understand this point, as the entire protein is degraded by the proteasome, so the degradation pattern for the upstream peptide should be identical to that for the degron peptide. It is concerning they are different. What is the pattern for other Dbn1 peptides across the 4 conditions?

Other:

1. Fig 1E: heat map suggests the lowest correlation coefficient between any two experiments (even between "no His" vs His Ub) is 0.94. Is this correct? If so, this emphasizes the high degree of non-specific protein enrichment (points 2 and 3 above)

2. Fig 3D: Cul4A DN inhibits Dbn1 degradation nearly as well as Cul1 DN. Authors should mention this in text and acknowledge there may be other Ub ligases that target Dbn1.

3. Fig 3J, K, L: TUBE pulldown in HeLa: missing ubiquitin blot of pulldown to show ubiquitin was pulled down equivalently among conditions

4. Fig S3D: MG262-treated lanes must be on same western membrane as non-MG262 lanes, in order to conclude that proteasome inhibition stabilizes ubiquitylated Dbn1 (text page 9)

5. Fig 3A: missing Dbn1 blot for lysate input

6. They should cite Ma et al. Mol Cell Prot 13:1659, 2014 in their Introduction as a tagged Sumo/ubiquitin-like approach previously used in Xenopus extracts to identify sumoylated proteins.

Reviewer #2 (Remarks to the Author):

In the manuscript entitled "Profiling ubiquitin signaling with UBIMAX reveals DNA damage-and SCFbTRCP-dependent ubiquitylation of the actin-organizing protein Dbn1", Colding-Christensen et

al. describe a method for the enrichment of ubiquitin-modified proteins and their characterization by mass spectrometry, UBIMAX. The authors applied this method to investigate DNA damage-induced ubiquitylation in the *Xenopus* egg extracts to identify a novel ubiquitylation target, Dbn1, and further elucidated the mechanism of Dbn1 regulation by phosphorylation, ubiquitylation, and proteasomal degradation in response to DNA double-strand breaks. To support the conclusions derived in the *Xenopus* egg extract system, the authors further conducted experiments in a human cancer cell line and performed an in-silico motif analysis of the human proteome indicating that the revealed mechanism might be conserved in different species. The manuscript is very well written, and the conclusions are well supported by the experiments presented in the study. However, I have a few minor suggestions, mostly in the proteomic analysis part.

- Page 6, paragraph 3, related to figure 1G: The description of the dynamic range might be wrong. The dynamic range, as a ratio of the maximal and minimal intensity measured is 7 on a log₁₀ scale, which I believe is ~10e7 fold, not ~7000 fold. Please, double check if this is correct.
- Page 7, paragraph 1, related to figure 1H: The text mentions that the 786 proteins were further interrogated, and 4 clusters were identified and shown in the heatmap. However, the heatmap already shows a subset of these 786 proteins. It should be clear in the text what are the proteins shown in the heatmap used for the clustering.
- Page 7, paragraph 1, related to figure 1H and S1H: The GO term enrichment analysis is based on a very small sample size which might give a significant result, but might be misleading. For example, figure S1H shows that the enrichment of the proteins related to inflammation in response to the addition of exogenous DNA was based on 2 proteins. The overrepresentation of the DNA repair proteins in the DSB-induced cluster seems to be more confident and based on more proteins. The authors might consider changing the description of their results in the part of describing the enrichment of the DNA-induced group or removing it. Furthermore, there are no terms related to inflammation in the Table S1, "Heatmap enrichment analysis" sheet, please double-check this supplementary file. Also consider matching the name of the clusters in this supplementary file to the heatmap and add a legend for this sheet to the Table S1 legend.
- Page 8, paragraph 1, related to figures 2D-I: If I understand it correctly, the data presented in the plots are based on two different MS experiments as shown in figure 1D and figure 2A. How did you analyze/process the intensity values from the two experiments, so the absolute values shown for DSB and ssDNA/SSB groups can be compared together?
- Page 12, paragraph 1, degnon motif: Have you found this motif in other proteins enriched in your initial experiment, e.g., in Ku80?
- Page 17, last paragraph: I think it would be more logical to include the in-silico motif analysis in the end of the results section rather than in the discussion.
- Page 38, Supplementary tables: Table S1 and Table S4 are not labelled in the sheets, and it was a bit difficult to understand which table is which. Furthermore, some of the supplementary tables are not called in the results section.

Reviewer #3 (Remarks to the Author):

In this manuscript by Colding-Christensen et al, a protein ubiquitination profiling method by mass spectrometry in *Xenopus* egg extract (names UBIMAX) is presented. The method relies on the addition of recombinant His-tagged ubiquitin to *Xenopus* egg extracts and the purification of ubiquitinated proteins under denaturing conditions, which ensures that ubiquitinated proteins are captured directly (i.e., ubiquitinated-protein to protein interactions are disrupted). As proof-of-concept, the authors focus on studying the DNA damage response when linearized plasmid DNA is added to *Xenopus* egg extracts (to simulate DNA double strand break (DSB) repair). This revealed several ubiquitinated proteins, including known DSB repair factors (e.g., Mre11, Nbn1, Rad50) and Dbn1, a protein that has not yet been implicated in DSB repair. The authors continue to characterise the mechanism of Dbn1 degradation and elegantly show that the cullin ring ligase (CRL) Cul1-beta-TRCP1 is responsible for degrading Dbn1 in response to DSBs. In the last part of the manuscript, the authors demonstrate that the degradation of Dbn1 is mediated by ATM and they identify the phospho-degnon.

The manuscript is well written and follows a clear flow of thoughts. The figures are informative and presented clearly. The experiments are well designed and include the correct controls. I

recommend publication of the present manuscript and I have a few suggestions for the authors on how to further strengthen it.

The authors perform all experiments by adding exogenous His-tagged ubiquitin to the *Xenopus* egg extracts. The authors include several controls to demonstrate that addition of recombinant ubiquitin does not induce any apparent artificial ubiquitination of target proteins. Nevertheless, a nice addition to this work would be to perform diGly peptide profiling as an orthogonal MS approach for quantifying ubiquitination sites in an endogenous setup.

The authors convincingly demonstrate that beta-TRCP1 is the Cul1 substrate receptor responsible for degrading Dbn1 in response to DSBs. I am wondering whether the authors investigated downstream consequences of Dbn1 depletion (or introduction of the non-degradable ATM phospho-deficient mutant)? For example, how is DSB repair affected when Dbn1 is depleted, or if cells cannot degrade Dbn1 in response to DSBs? Is cancer cell survival affected by non-degradable (or hyper-phosphorylated) Dbn1 mutants? Since Dbn1 is an actin cytoskeleton organizing protein, are there consequences for cell division when Dbn1 is depleted? And are there any known cancer mutations in either Dbn1 or beta-TRCP1? It would be helpful, if the authors could discuss these points.

Figure 1E: Besides showing quantitative correlations, it would be interesting to see how precise the quantification of the detected proteins is.

Figure S1G: Indicate how many proteins were identified in how many conditions (to get a better understanding on the data completeness/number of missing values)

Figure 3C: The proof that dominant negative Cul1 supplementation to the egg extracts rescues Dbn1 degradation looks convincing; however, I noticed that also Cul4a seems to rescue its degradation. Could the authors comment on this? Is it possible that more than one CRL complex regulate Dbn1 protein stability?

Point-by-Point response to reviewers

General remarks

We appreciate the time and expertise of the reviewers in evaluating our manuscript. Their insightful comments and suggestions have provided valuable perspectives that have enhanced the quality of our work. In this rebuttal letter, we address each of the raised points, highlighting the modifications made to strengthen the manuscript. As a result, we are confident that the revised version of our manuscript is greatly improved.

REVIEWER COMMENTS

Reviewer #1 (Remarks to the Author):

This paper by Colding-Christensen et al. describes a new technique (UBIMAX) to identify proteins whose ubiquitylation is induced by DNA damage in *Xenopus* oocyte extracts. Extracts supplemented with recombinant His-Ub are incubated with a defined DNA lesion, and ubiquitylated proteins are enriched by denaturing pulldown followed by mass spectrometry. Using this technique, the authors find that the actin binding protein Dbn1 is ubiquitylated by Cul1- β Trcp in response to double strand breaks (DSBs) at an ATM-stimulated degron to trigger proteasomal degradation of Dbn1.

The paper does a nice job characterizing DSB-induced ubiquitylation of Dbn1 by Cul1- β TRCP and showing the ATM-regulated degron is sufficient for inducing ubiquitylation of a heterologous protein. The mass spectrometry data is nicely controlled (with “no His” and E1 inhibitor conditions). The DSB experiments are performed in quadruplicate with good correlation, and the data are analyzed well. However, like prior tagged Ub approaches, the technique does not identify many ubiquitylated proteins (see point 1), does not identify sites of ubiquitylation, and enriches nonspecific proteins. Also, there is no functional characterization of Dbn1 ubiquitylation in DNA repair.

We would like to thank the reviewer for their endorsement of our manuscript and for recognizing that our experiments adhere to the rigorous standards mandated for publication in Nature Communications.

We have addressed the reviewer’s comments in separate points below.

1. The authors identify a total of only 786 ubiquitylated proteins from their quadruplicate DSB experiments (text page 6, Fig S1G). With an FDR <0.05, they find only 23 proteins induced by DSBs (Figs 1H, S1G, S1I; Table S1). With a more stringent FDR <0.01, they find 7 proteins (Fig S2B, Table S1). Similarly, ssDNA-DPC (Figs 2B, S2B) induces only 19 proteins (at FDR < 0.05) and 8 proteins (at FDR < 0.01), while SSB-DPC (Figs 2C, S2B) induces 19 proteins (at FDR < 0.05) and 8 proteins (at FDR < 0.01). Newer techniques like diGly peptide enrichment or UbiSite (Akimov NSMB 2018) were developed to identify ubiquitylated proteins more efficiently than tagged ubiquitin/UBL approaches (Meierhofer J Prot Res 2008; Danielsen Mol Cell Prot 2011; Ma et al. Hunter Mol Cell Prot 2014). This is a drawback of their approach.

We agree with the reviewer's notion that other proteomics methods exist that allow comprehensive identification of ubiquitylation sites, such as the DiGly and UbiSite strategies^{1,2}. While these methods have allowed gaining new insights into the functional consequences of ubiquitylation, we would respectfully like to point out that the aim of our manuscript was never to break records with regard to the number of identified ubiquitylated proteins.

Moreover, strategies towards global characterization of ubiquitylation sites remain limited in their ability to determine the dynamic and quantitative regulation of the ubiquitylated protein targets, and the extent of ubiquitylation events elucidated by these approaches render them impractical for deciphering highly specific responses to specific stimulations and/or conditions. Hence, we developed the UBIMAX method, which combines advances in proteomics technologies with *Xenopus laevis* egg extracts, to dissect ubiquitylation signalling pathways under specific stresses with high temporal resolution.

While the number of ubiquitylated proteins in our manuscript may be viewed by the reviewer as a limitation, we would like to draw the reviewer's attention to the fact that the DiGly and UbiSite methods are fundamentally different strategies that necessitate the utilization of substantial amounts of starting material to achieve their reported number of ubiquitylation sites. For example, in the hallmark paper by the Gygi group², the authors used 25-35 milligrams of protein per sample. In contrast, the UBIMAX experiments necessitate only approximately 300 micrograms of protein material per sample, which represents a reduction of two orders of magnitude in starting material. This equates to 6.25 microliters of *Xenopus* HSS extract, amounting to less than 1/50th of the total extract material yielded from a single frog. As a result, all experiments detailed in this manuscript can be conducted employing egg extract obtained from a single frog, underscoring the practicality of the UBIMAX approach for laboratories utilizing *Xenopus* extracts and at the same time considering the necessary use of experimental animals. In comparison, the DiGly and UbiSite methods would demand egg extracts sourced from 40 frogs, rendering such endeavours considerably unfeasible.

To clarify this, we have added the following sentence in the discussion (page 16, 2nd paragraph):

"UBIMAX does not identify a vast amount of ubiquitylated proteins when compared to other proteomic strategies which identify site-level ubiquitylation¹³⁻¹⁷ but does identify dynamic protein ubiquitylation in response to specific stimuli while expending 100-fold less starting material."

Given the reviewer agrees that the presented work is of high technical quality, we believe that UBIMAX constitutes an important methodological advance and very useful for the broad readership of Nature Communications.

2. In Fig S1F, enriched peptide abundance for the E1 inhibitor and "no His" controls (bars 4 and 5) is comparable to that for the His-Ub enrichment conditions (bars 1-3). This suggests the majority of peptides purified by Ni-NTA beads are binding non-specifically, which highlights a limitation of their approach.

We acknowledge the reviewer's analysis of this data. Indeed, our UBIMAX approach does entail a certain level of background-binding proteins. However, we would like to point out that background-binding proteins are an inherent challenge of essentially all proteomics experiments and not unique to our UBIMAX approach.

Furthering this, we would like to highlight that even global proteomic studies, as mentioned in the previous comment (DiGly and UbiSite), also entail a high degree of background-binding proteins. To illustrate this, we examined the proportion of identified peptides that corresponded to ubiquitylated peptides as opposed to unmodified peptides as an indicator of 'undesired' or 'background-binding proteins' in studies employing the

DiGly method (Pride database entries PXD030714 and PXD024103). From this analysis, it emerged that only 25-35% of the total identified peptides were associated with ubiquitylated species. Similarly, for UbiSite analysis (Pride database entries PXD006201¹ and PXD022367³), the corresponding percentages revealed that merely 20% constituted ubiquitylated peptides.

Illustrating that background-binding proteins represent a common phenomenon not limited to our UBIMAX approach, this underscores the importance of including proper controls to reliably test for true interactors⁴.

As acknowledged by the reviewer, our UBIMAX strategy encompasses these pertinent and meticulous controls, enabling us to effectively distinguish between background-binding proteins and ubiquitylated targets in a quantitative manner.

3. In Fig 1F, E1 inhibition reduces the percentage of ubiquitin peptides enriched on nickel beads, which again suggests a high prevalence of non-specific binding to beads. If the majority of peptides are enriched via specific pulldown by covalent attachment to His-Ub, E1 inhibition should increase the ratio of ubiquitin peptides to non-ubiquitin peptides (by decreasing covalent attachment of His-Ub to other non-ubiquitin proteins), but this does not occur.

We understand the reviewer's point that under ideal conditions of the biological system employed, and sample preparation utilized, the ubiquitin E1 inhibitor condition should allow specific enrichment of His-ubiquitin. However, we would respectfully like to point out that no biological system nor sample preparation strategy is without variance or background.

For example, following the reviewer's notion, treatment with the ubiquitin E1 inhibitor should enrich unconjugated His₆-ubiquitin, whereas the "no DNA", "DNA", and "DSB" conditions would likely result in an enrichment of both mono- and poly-ubiquitylated forms. This alteration in the ubiquitin composition across samples impacts the overall ubiquitin percentage, with poly-ubiquitinated forms providing a stronger MS signal (i.e. higher ubiquitin percentage in the sample) as compared to mono-ubiquitylation or un-conjugated His₆-ubiquitin. Beyond this, the effectiveness of the ubiquitin E1 inhibitor may not be 100%, which may lead to detection of some ubiquitylated proteins in the replicates treated with the inhibitor. Collectively, the above will skew the reviewer's argument to some degree.

Expanding on this, we examined the overall ubiquitin levels in standard His₆-ubiquitin enriched samples as opposed to ubiquitin E1 inhibitor-treated and "no His" control samples (Rebuttal Fig. 1). This analysis revealed a much more pronounced ubiquitin presence in the His₆-Ubiquitin enriched samples when contrasted with those subjected to ubiquitin E1 inhibitor treatment.

Rebuttal Fig. 1. In the above figure, mean values are graphed with error bars representing standard deviations.

This is likely due to the effects mentioned above along with the notion that in His₆-Ubiquitin enriched samples, UBIMAX co-enriches both exogenous His₆-Ubiquitin and endogenous ubiquitin, as both will be present in the samples and likely are co-conjugated on target proteins.

Based on this, we have argued in the manuscript that the decrease in ubiquitin signal from ubiquitin-enriched samples to ubiquitin E1 inhibitor-treated samples indicates the expected loss of enrichment of endogenous ubiquitin with ubiquitin E1 inhibition and *vice versa* the gain of conjugated endogenous ubiquitin on top of His₆-ubiquitin in the ubiquitin-enriched samples compared to enrichment of mainly un-conjugated His₆-ubiquitin in the ubiquitin E1 inhibitor samples.

Still, as outlined by the reviewer, effects derived from non-specific binding across the samples will likely contribute to the observations. However, non-specific binding of proteins (i.e. background-binding proteins) constitute an inherent obstacle in all proteomic experiments – particularly in enrichment-based experiments. Hence, we would like to emphasize that non-specific binding of proteins to beads is not exclusive to the UBIMAX approach.

In summary, we acknowledge that our method entails non-specific binding among the investigated samples, similar to other proteomics experiments. Nevertheless, we anticipate that by using proper quantitative measures and statistical filtering, these non-specific binding effects will not affect the validity of our conclusions.

Still, we have now included a sentence in the revised manuscript to address the aspects of non-specific binding (page 5, 4th paragraph), which reads:

“However, as background binding proteins are an inherent challenge in any enrichment-based proteomic experiment³⁵, we utilized label free quantification (LFQ)³⁰ of replicate samples to be able to distinguish background binding proteins from true ubiquitin target proteins.”

4. They should perform UBIMAX in extracts treated with a proteasome inhibitor. This will increase sensitivity for ubiquitylation events that cause degradation by the proteasome

We thank the reviewer for this suggestion and agree that performing UBIMAX on extract reactions treated with proteasome inhibitor could allow for increased quantitative regulation of proteins that become ubiquitylated and degraded by the proteasome in response to DSBs.

In response to the reviewer's suggestion, we have performed UBIMAX in extracts stimulated for 30 minutes with undamaged DNA or DSB plasmid DNA in the presence or absence of proteasome inhibitor (MG262), with all conditions performed in quadruplicates. In support of an effective proteasomal treatment, we noted that addition of proteasome inhibitor to extract increased the prevalence of high molecular weight ubiquitin-conjugates (Rebuttal Fig. 2a). However, we also noted that proteasome inhibition quickly depleted the free ubiquitin pool. This will likely impair induction of further ubiquitylation events in response to any stimulus applied, and potentially introduce pleiotropic effects in the readout compared to our initial experiments.

Still, this UBIMAX experiment (performed without "noHis" or ubiquitin E1 inhibitor background controls) confirms almost all the DSB-induced ubiquitylation events reported in our manuscript (Fig. 1i) and additionally detects 58 proteins for which ubiquitylation is upregulated only by the combination of MG262 and DSBs (Rebuttal Fig. 2b and coloured purple in Rebuttal Fig. 2d). These 58 proteins are candidates for DSB-induced and ubiquitin-mediated proteasomal degradation within the first 30 min of stimulation with DSBs. These data confirm the reviewer's notion, that the sensitivity of UBIMAX may be increased by performing the experiment in the presence of a proteasome inhibitor.

Importantly for the conclusions of this manuscript, and in agreement with the time course western blot analyses presented herein, our UBIMAX experiment using MG262 confirms that Dbn1 is ubiquitylated upon DSBs but not yet significantly degraded by the proteasome within 30 min of DSB stimulation (Dbn1 marked in black in Rebuttal Fig. 2d). Collectively, besides confirming our initial data described in the manuscript, the experiment outlined here furthers the value of UBIMAX in performing varied analyses tailored precisely to the scientific aim in question.

However, because proteasomal inhibition has an indirect effect on ubiquitin dynamics, we do not feel confident adding this data to the manuscript but add it here for the reviewer's perusal.

Rebuttal Fig. 2. a. Western blot analysis of UBIMAX DSB-MG262 experiment. **b.** Venn diagram and tables detailing proteins showing increased ubiquitylation upon DSB (red) or DSB+MG262 (purple) treatment and subsequent UBIMAX analysis. **c-d.** Volcano plot analysis comparing ubiquitylated proteins enriched from DSB versus undamaged DNA-treated samples (c) or DSB+MG262 versus DSB-treated samples (d). Blue, red and purple dots indicate significantly enriched ubiquitylated proteins with undamaged DNA, DSBs and DSBs+MG262, respectively. Significance was determined by two-tailed Student's t test, with permutation-based FDR-control with $S_0 = 0.1$ and 2500 rounds of randomization, to ensure $FDR \leq 0.05$.

5. The connection of DNA damage to Dbn1 degradation is interesting given recent results showing that actin regulates DSB localization, resection, and repair (Belin eLife 2015, Schrank Nature 2018, Caridi Nature 2018). The authors should test whether Dbn1 depletion (and importantly addback of S609A Dbn1) affects DSB repair. They should check their NHEJ assay (Fig S1B-C) where they have shown E1 inhibition inhibits repair. If no phenotype is found there, they should look for other evidence of defects in DSB repair. They could check DSB resection (Liao NAR 2012) or resection/recombination-based repair assays in *Xenopus* (see references cited in Graham, Meth Enzym 2017). Given that the Dbn1 ubiquitylation event is conserved in

HeLa cells, they could also check HR, resection, RAD51 foci formation, or other repair assays in mammalian cells.

We would like to thank the reviewer for these very insightful suggestions on how to investigate the effect of Dbn1 ubiquitylation on DSB repair.

Following the reviewer's suggestion, we have explored the role of Dbn1 ubiquitylation in the DSB response in more detail. Initially, as suggested by the reviewer, we conducted experiments to investigate the impact of Dbn1 depletion and reintroduction of the Dbn1-S609A mutant on DSB repair within our non-homologous end joining (NHEJ) assay (with depletion of Ku80 as a control, Rebuttal Fig. 3). Regrettably, we observed no substantial effect in this assay.

Rebuttal Fig. 3. a. DNA repair assay using mock-, Dbn1-, or Ku80-depleted extracts complemented with the indicated recombinant proteins. The experiment was conducted in triplicate, yielding consistent outcomes. A representative experiment is shown. **b-c.** Quantification of percent remaining DSB substrate (linear plasmid DNA, b) and appearance of repaired product (supercoiled (SC), open circular (OC), and dimer, c).

Next, we wondered whether we might be able to detect an effect on resection as a means of indicating an effect on DSB repair. To this end, we assayed resection in a similar reaction to the above in mock- or Dbn1-depleted extracts. As resection is limited in HSS, we performed this experiment in both HSS and nucleoplasmic extracts (NPE), which is a resection/HR competent extract⁵. Again, we observed no significant effect when we deplete Dbn1 (Rebuttal Fig. 4).

Rebuttal Fig. 4. DSB processing was analysed by DNA repair assay in mock- or Dbn1-depleted HSS or NPE. The experiment was conducted in triplicate. A representative experiment is shown.

As Actin-binding proteins associate with DSBs, and actin filament polymerization was proposed to regulate DSB localization and repair by homologous recombination (HR) specifically⁶⁻¹⁰, we decided to assay a potential effect of Dbn1 depletion and add back on HR. Furthermore, if the role of Dbn1 in DSB repair is performed via actin filament regulation, the lack of effect observed in egg extracts may be due to the fact that to maintain a soluble environment, our egg extracts are supplemented with actin polymerization inhibitors and may therefore not be a suitable system to investigate the function of Dbn1. To circumvent this, we moved into human cells and took advantage of the DR-GFP human cell line, in which a site-specific DSB is induced in a GFP cassette by induction of I-SceI endonuclease expression. As this cassette can only be repaired in a manner restoring GFP signal by HR-mediated gene conversion of a donor cassette, GFP signal serves as a read-out for HR activity. We performed this assay in cells either mock-transfected or with siRNA-mediated knock down of DBN1 either with or without complementation with WT or S599A mutant DBN1 by transient plasmid transfection (Rebuttal Fig. 5). We observed no reproducible effect with this assay either.

Rebuttal Fig. 5. DR-GFP cell line-based HR assay performed in cells transfected with control siRNA or siRNAs targeting *BRCA2* or the UTR of *DBN1* as well as with expression plasmids encoding either WT or S599A mutant *DBN1* as indicated. The experiment was conducted in duplicate. A representative experiment is shown.

As suggested by the reviewer, we also explored whether DBN1 status might affect DSB repair in human cells by influencing either DNA damage-detection/signalling or termination of DNA repair and DNA damage signalling. To address this, we assayed γ H2AX foci appearance and clearance after 10 Gy ionizing radiation (IR) in cells with either mock- or DBN1 siRNA-mediated knockdown (Rebuttal Fig. 6a). To further assess a potential effect of introducing the non-degradable DBN1 S599A mutant, we also generated human cell lines either WT or CRISPR-mediated knock-out (KO) for DBN1 as well as the DBN1 KO cell line complemented with stable Flp-In integration of doxycycline (dox) inducible WT or S599A mutant DBN1 (Rebuttal Fig. 6b). Using these cell

lines, we assayed γ H2AX foci 30 minutes after IR (Rebuttal Fig. 6c-d). As seen from the data of these two experiments, there is a tendency towards a decrease in the appearance of γ H2AX foci with loss of DBN1. Nonetheless, the observed impairment is comparably rescued by both WT DBN1 and the S599A DBN1 mutant, suggesting that potential deficiencies in DNA damage signalling resulting from loss of DBN1 could encompass diverse factors beyond the ubiquitylation state of DBN1.

Rebuttal Fig. 6. **a.** Distribution of number of γ H2AX foci in human cells transfected with control- or three different siRNAs targeting *DBN1* and analysed by immunofluorescence microscopy at the indicated timepoints following 10 Gy IR. **b.** Distribution of DBN1 abundance analysed by immunofluorescence microscopy in human cell lines either parental, CRISPR *DBN1* KO, or the latter complemented with stable inducible expression of WT or DBN1-S599A mutant. Lines indicate the median. Y-axis was capped at 500, excluding 78 out of 5642 data points. a.u., arbitrary units. **c.** Human cells either parental, CRISPR *DBN1* KO, or the latter complemented with stable inducible expression of WT or DBN1-S599A mutant were either mock-treated or exposed to 10 Gy IR and analysed as in a 30 min post-IR. **d.** Percent of cells in the experiment detailed in **c** showing more than 10 γ H2AX foci. For **a** and **c**, full lines indicate median values, while dotted lines indicate 25% and 75% quartiles.

Finally, we assayed cell survival in response to various DSB-inducing treatments using the Incucyte live cell analyser. Employing the identical cell lines as previously mentioned, we assayed cell growth and survival by measuring confluency for five days in the absence or presence of the DSB-inducing agents Etoposide and CPT (Rebuttal Fig. 7a-b).

Rebuttal Fig. 7. a-b. Incucyte-based survival assay measuring cell confluency with time in either parental, CRISPR *DBN1* KO, or the latter complemented with stable inducible expression of WT or *DBN1*-S599A mutant in response to either mock-, 500 nM etoposide- (a), or 20 mM CPT treatment (b). The concentrations used were determined by prior titration of parental cells. These experiments were done in technical (cell culture) triplicates. Displayed are the mean values with error bars representing standard deviations.

To assay the effect of IR, we additionally performed colony formation assays in response to 2 Gy IR (the survival of parental cells was too compromised to generate any meaningful comparisons of survival at 10 Gy IR, which we have otherwise used for short term experiments, Rebuttal Fig. 8).

Rebuttal Fig. 8. Survival measured by colony formation of either parental-, CRISPR *DBN1* KO cells, or the latter complemented with stable inducible expression of WT or *DBN1*-S599A mutant in response to 2 Gy IR. Mean values are graphed with error bars representing standard deviations. Significance was determined by one-way ANOVA with Dunnett's multiple comparisons test for all conditions against the parental control. All comparisons produced non-significant p-values.

Based on the collective outcome of these survival experiments, we conclude that *DBN1* status does not significantly affect cell viability in response to DSBs.

We suspect that the lack of observable effects in the above-described experiments may stem from only a specific pool of the *DBN1* protein being ubiquitylated in response to DSBs. To further this, we performed cell synchronization experiments and found that IR-induced ubiquitylation of *DBN1* mainly occurs in G1-phase in human cells. We consider this new data on the cell cycle regulated nature of *DBN1* ubiquitylation relevant to the current manuscript and have added it to the revised manuscript (new Supplementary Fig. 3n).

Furthermore, as DSB-induced DBN1 ubiquitylation requires prior ATM-mediated phosphorylation, this further suggests that only a potential nuclear pool of DBN1 is targeted for ubiquitylation. DBN1 is mainly a cytosolic protein but could be imported into the nucleus similarly to actin.

Finally, the data presented here indicates that there is no major effect of DBN1 ubiquitylation status on DSB repair. We believe that DBN1 ubiquitylation may represent a more subtle regulation of DSB repair, e.g. through regulation of actin-mediated movement of DSBs^{6,8-10}. Hence, uncovering a clear mechanistic effect of DBN1 ubiquitylation status on DSB repair, would entail e.g. monitoring DSB movement while perturbing DBN1 ubiquitylation and degradation – which would be time-wise cumbersome as well as experimentally and technically outside of our expertise.

While we fully acknowledge the interesting aspect of further characterizing the functional implications of DBN1 ubiquitylation on DSB repair, we believe this would lie outside the scope of the current manuscript, which is primarily aimed at presenting the creation of a proteomic method designed to unveil novel ubiquitylated targets that could be pursued in separate investigations. In addition, the findings of our manuscript are the first to show an actin-related protein being regulated directly by DSBs and we feel that this would be of great interest to the actin- and DSB repair field.

6. Along these lines, it has been shown that actin and actin regulators localize to chromosomal DSBs in *Xenopus* extracts (Schrank Nature 2018). They should test whether Dbn1 is recruited to chromosomal DSBs (like Schrank et al) or to their DSB plasmid (as they do in Larsen, Mol Cell 2019)

We thank the reviewer for the suggestion to investigate Dbn1 localization to DSB DNA. Following this, we have assayed for recruitment of Dbn1 to our linearized DSB plasmid via plasmid pulldown experiments (as done in¹¹). However, while we readily detect recruitment and ubiquitylation of Ku80 on DSB plasmid DNA, we do not detect any recruitment of Dbn1. This data indicates that, in contrast to Ku80, Dbn1 does not localize to DSBs nor does DSB-induced ubiquitylation of Dbn1 occur on DNA. We find this experiment to be conclusive and have therefore added it to the revised manuscript (new Supplementary Fig. 3f).

7. Fig 3l: they need to blot for Cul1 and Skp1 to determine the extent to which β Trcp immunodepletion reduces the levels of Cul1 and Skp1. To show specificity, they add back β Trcp, but β Trcp is translated in reticulocyte lysates which may contain Cul1 and Skp1.

In reference to the reviewer's request, we have blotted for Cul1 in the experiment shown in old Supplementary Fig. 3h / new Supplementary Fig. 3j, confirming that a major pool of Cul1 remains in the extract following β -Trcp1 immunodepletion.

To assay whether a minor (but functional) co-depletion of Cul1 were to occur with β -Trcp1 immunodepletion, we have furthermore blotted for Ku80 in the same experiment. Ku80 is a known ubiquitylation target of the Cul1-containing SCF ubiquitin ligase upon DSBs, although recognized by a different F-box protein, Fbxl12^{12,13}. As β -Trcp1 immunodepletion does not impair Ku80 ubiquitylation in response to DSBs, we conclude that we do not functionally co-deplete Cul1 with β -Trcp1.

We have replaced the panel in old Supplementary Fig. 3h / new Supplementary Fig. 3j with this extended experiment in the revised manuscript.

8. The authors should explain why they observe conflicting results for Dbn1 and Ku80 degradation in different figures. This must be reconciled.

a. DSB induces Ku80 degradation at 120 min in Fig S3E but not in Fig 3D.

b. They see DSB-induced WT Dbn1 degradation in Fig 4C but not in Fig S4B. In both cases of these cases, Dpn1 was depleted and then WT Dbn1 was added back

We apologize for the conflicting results related to Dbn1 and Ku80, and we have therefore replicated the experiments with minor changes to the experimental protocol as detailed below.

a. Because reaction kinetics can vary between batches of egg extract, differences in kinetics of ubiquitylation and degradation can sometimes be observed in experiments conducted with different extract preparations. We have repeated the experiment shown in Fig. 3d using a new extract batch, which shows induction of Ku80 ubiquitylation upon DSB addition and concomitant reduction of the unmodified band at 120 minutes in accordance with Supplementary Fig. 3e. We have replaced the panel in Fig. 3d with this new experiment in the revised manuscript.

b. As the sole experiment in this study, the experiment shown in Supplementary Fig. 4b was done using *in vitro* translated Dbn1 proteins produced in wheat germ extract rather than in reticulocyte extract. We suspect that this may have resulted in a differently processed protein. We have redone the experiment with batches of Dbn1 proteins produced in reticulocyte extracts. In accordance with Fig. 4c, this new experiment shows degradation of WT Dbn1 with DSBs in the untreated condition (lanes 1-5). We have replaced the panel in Supplementary Fig. 4b with this new experiment in the revised manuscript.

9. Fig 3G, Table S4: why is ATMi decreasing the majority of Dbn1 interactions (with proteins like actin and the actin regulators capza1, capzb, arpc3a, arpc5, arpc2b). It makes sense that ATMi would block interaction with β Trcp, Skp1, and Cul1, but I would expect ATMi to increase basal interaction with actin (and its binding partners) because ATMi should stabilize Dbn1. Also, their MS data in Table S4 (corresponding to Fig 3G) show that Dbn1 levels are not higher upon ATM inhibition. These results suggest the ATMi IP sample may have overall lower protein/extract input, and the authors should address this.

We respectfully disagree with the reviewer's interpretation that our data indicates that the ATMi IP samples may have lower protein/extract input:

1. Although the reviewer is right in their reasoning that ATMi blocks Dbn1 ubiquitylation/degradation, we want to emphasize that Dbn1 degradation occurs with relatively slow kinetics and that it takes more than 60 min before an observable impact on the Dbn1 protein pool is evident. We do not know why this is, but we suspect the involvement of phosphatases counteracting Dbn1 phosphorylation. However, as we perform the Dbn1 IP after 60 minutes of incubation with DSB plasmid in this experiment, before the major pool of Dbn1 is degraded, we do not expect a drastic effect of ATMi on Dbn1 stability at the time point sampled.
2. As Dbn1 is the target of our immunoprecipitation approach in this IP-MS experiment, the amount of Dbn1 protein in the different samples will not necessarily correspond to the total amount of Dbn1 protein in the extract reactions from which we have sampled. We believe that the fact that there is no statistical difference in the amount of Dbn1 protein between the Dbn1 IP samples rather proves that we have immunoprecipitated similar amounts across the samples and thus serves as a good immunoprecipitation control.

For the above reasons, we therefore do not expect that ATMi would have a profound impact on the Dbn1 interactome, except in the case of Dbn1 interactors that directly depend on Dbn1 phosphorylation status (e.g. the SCF ^{β -Trcp1} complex). We would also like to note that some of the actin-related Dbn1 interaction partners are unchanged by the ATMi condition (e.g. Acta1 and Arpc4). This emphasizes the fact that while ATM may regulate some interactions between Dbn1 and actin-related and -regulating proteins, other interactions seem to be unaffected.

To reconcile this point, we have graphed the summed peptide intensity across the samples of this experiment, which similarly shows that the overall protein/extract input is similar across all the Dbn1 IP samples. We have added this panel to the revised manuscript (new Supplementary Fig. 3g).

10. Fig S3H, Fig 3I: why are basal levels of Dbn1 (at 1 min) lower in β Trcp-depleted extracts? It is nice how degradation is inhibited (i.e., Dbn1 levels are stable over time), but why do Dbn1 levels start out lower? This becomes an issue in Figs S3I/J, where it looks like β Trcp depletion is abolishing Dbn1 ubiquitylation but not increasing Dbn1 levels. This may result from loss of Dbn1 from extracts by immunodepletion of β Trcp. If so, this would suggest β Trcp binds Dbn1 in the absence of DNA damage. The authors should address this observation and implications in the text.

We completely agree with the reviewer's notion that β -Trcp1 immunodepletion co-depletes a pool of Dbn1 (as seen from the figures mentioned by the reviewer).

As further suggested by the reviewer, this may indicate that Dbn1 interacts with β -Trcp1 in the absence of DNA damage. This may be due to the fact that *Xenopus* egg extracts are "living" entities that, just as cells, are always subjected to endogenous DNA damage and or/basal activation of the DDR kinases. This may be the reason why we observe a pool of Dbn1 interacting with β -Trcp1 in the absence of addition of exogenous DNA damage.

Alternatively, this may be due to a transient interaction through the non- or mono-phosphorylated degron, but that the interaction only becomes productive (i.e. induces Dbn1 ubiquitylation) upon full phosphorylation of the degron motif^{14,15}.

In any case, our Dbn1 IP-MS (Fig. 3g-h) shows that the interaction between Dbn1 and β -Trcp1 is greatly increased upon DSBs compared to unstimulated (undamaged) conditions which is in perfect agreement with our thorough characterization of the Dbn1 β -Trcp1 degron.

We have added the following comment regarding this to the main text of the revised manuscript (page 11, 3rd paragraph):

"The minor co-depletion of Dbn1 with β -Trcp1 immunodepletion suggests an interaction in unperturbed conditions, which is significantly further induced by DSBs (Supplementary Fig. 3j and Fig. 3g-h)."

11. Fig 3J, K, L: why is IR not inducing Dbn1 degradation in HeLa cells? If the authors believe Dbn1 ubiquitylation serves a different purpose in HeLa, they should acknowledge and address this in the manuscript.

As described above in connection to point #5, we believe that only a fraction of DBN1 (i.e. nuclear DBN1 in G1 cells) is targeted for ubiquitylation and subsequent degradation in response to DSBs. Hence, the

visualization of the degradation of this particular subset through western blot analysis of whole cell extracts presents a challenging endeavour.

We have acknowledged this aspect in the revised manuscript (page 12, 1st paragraph), which should alleviate the reviewer's concerns:

"However, concomitant reduction of unmodified DBN1 was not apparent in the input, suggesting that a minor pool of DBN1 protein is targeted for ubiquitylation upon IR-induced DNA damage in human cells."

12. Fig S4A, text on p.12: "lack of detection was not due to the general Dbn1 sequence context, as the upstream peptide was detected equally across all conditions." I do not understand this point, as the entire protein is degraded by the proteasome, so the degradation pattern for the upstream peptide should be identical to that for the degron peptide. It is concerning they are different. What is the pattern for other Dbn1 peptides across the 4 conditions?

We would like to apologize for any confusion this section may have caused. As outlined in our response to the reviewer's point #9, we perform Dbn1 immunoprecipitation following 60 minutes incubation with DSB plasmid. At this timepoint, the major pool of Dbn1 has not yet been degraded. Additionally, we would like to point out that as Dbn1 is the immunoprecipitation target in this experiment, the amount of individual Dbn1 peptides in the different samples will not always reflect the total amount of Dbn1 protein in the extract reactions.

Furthermore, as the Dbn1 protein exist in various modified variants (i.e. proteoforms), certain peptides containing modifications may only be present in smaller amounts, or only present in some samples, while 'common' peptides shared across all proteoforms will be more abundantly present across all samples. To illustrate this and following the reviewer's suggestion, we have analysed all the Dbn1 peptides detected in this experiment across the four conditions (Rebuttal Fig. 9). This shows that indeed most Dbn1 peptides do not significantly change across the conditions, except for the peptide containing the β -Tcrp1 degron and a handful of others. While it would be interesting to analyse the contribution of potential sites of modification on these Dbn1 peptides in more detail, we believe such investigations are beyond the scope of the current manuscript.

Rebuttal Fig. 9. Heatmap showing log₂-transformed Dbn1 peptide abundances originating from the Dbn1 IP-MS experiment. Peptides are ranked by q-value as determined by ANOVA with permutation-based FDR. Peptides for which FDR < 1% are shown in blue.

Other:

1. Fig 1E: heat map suggests the lowest correlation coefficient between any two experiments (even between “no His” vs His Ub) is 0.94. Is this correct? If so, this emphasizes the high degree of non-specific protein enrichment (points 2 and 3 above)

As outlined in our comments to points #2 and #3 above, we agree with the reviewer’s notion that our UBIMAX approach entails enrichment of non-specific proteins to an extent similar to any other enrichment-

based proteomics experiment. As non-specific proteins will be highly reproducibly detected across similar samples, these will contribute to the linear relationship between variables. However, as different Pearson correlation coefficients are detected between replicates of control and biological conditions, these differences indicate the biologically relevant differences between these conditions.

As for the Pearson correlations reported in Fig. 1e, these are calculated using the Perseus software and are, as according to this calculation, correctly reported.

2. Fig 3D: Cul4A DN inhibits Dbn1 degradation nearly as well as Cul1 DN. Authors should mention this in text and acknowledge there may be other Ub ligases that target Dbn1.

We thank the reviewer for pointing this out. We have since realised that this experiment was done with different concentrations of CulDN proteins. Specifically, the concentration of Cul4a DN unfortunately turned out to be almost 4-fold higher than that of Cul1 DN (1.1 $\mu\text{g}/\mu\text{l}$ and 0.3 $\mu\text{g}/\mu\text{l}$, respectively).

To test whether the effect of Cul4a DN was due to this higher concentration, we performed a titration experiment (Rebuttal Fig. 10). This experiment shows that Cul4a DN does not inhibit Dbn1 degradation at lower concentrations comparable to those of Cul1 DN (0.2 $\mu\text{g}/\mu\text{l}$), which do. We have now redone the experiment shown in Fig. 3d with titrated amounts of Cullin DN proteins (all at 0.2 $\mu\text{g}/\mu\text{l}$) and replaced the panel in the revised manuscript.

Rebuttal Fig. 10. Dbn1 ubiquitylation was assayed by western blot analysis in the presence of DSB plasmid DNA and different amounts of Cul1 or Cul4a dominant negative recombinant proteins.

However, the reviewer is correct that we cannot exclude that other ubiquitin ligases (not assayed for in this study) may target Dbn1 and we have now acknowledged this in the revised manuscript (page 10, 3rd paragraph):

“While we cannot exclude a potential contribution of other ubiquitin ligases not assayed for, we further validated Cul1-dependent ubiquitylation of Dbn1 by immunodepletion of Cul1 from egg extracts.”

3. Fig 3J, K, L: TUBE pulldown in HeLa: missing ubiquitin blot of pulldown to show ubiquitin was pulled down equivalently among conditions

We have now included ubiquitin blots of the pulldown samples in Fig. 3j-l and replaced these figures in the revised manuscript.

4. Fig S3D: MG262-treated lanes must be on same western membrane as non-MG262 lanes, in order to conclude that proteasome inhibition stabilizes ubiquitylated Dbn1 (text page 9)

The eight samples shown in this figure are in fact on the same membrane. They are shown separated because they were not juxtaposed on the membrane. We have added this information to the legend of Supplementary Fig. 2d.

5. Fig 3A: missing Dbn1 blot for lysate input

The input control for this experiment is shown in Supplementary Fig. 3a. We did not sample the reactions for total extract input throughout the time course of this experiment. However, the ubiquitin blot shown in Fig. 3a serves as a pulldown control, which we believe is more relevant control in this context.

6. They should cite Ma et al. Mol Cell Prot 13:1659, 2014 in their Introduction as a tagged Sumo/ubiquitin-like approach previously used in Xenopus extracts to identify sumoylated proteins.

We thank the reviewer for bringing this paper to our attention. We have now included this reference in the introduction of the revised manuscript (page 3, 2nd paragraph, Ref. 23).

Reviewer #2 (Remarks to the Author):

In the manuscript entitled “Profiling ubiquitin signaling with UBIMAX reveals DNA damage-and SCFbTRCP-dependent ubiquitylation of the actin-organizing protein Dbn1”, Colding-Christensen et al. describe a method for the enrichment of ubiquitin-modified proteins and their characterization by mass spectrometry, UBIMAX. The authors applied this method to investigate DNA damage-induced ubiquitylation in the Xenopus egg extracts to identify a novel ubiquitylation target, Dbn1, and further elucidated the mechanism of Dbn1 regulation by phosphorylation, ubiquitylation, and proteasomal degradation in response to DNA double-strand breaks. To support the conclusions derived in the Xenopus egg extract system, the authors further conducted experiments in a human cancer cell line and performed an in-silico motif analysis of the human proteome indicating that the revealed mechanism might be conserved in different species. The manuscript is very well written, and the conclusions are well supported by the experiments presented in the study. However, I have a few minor suggestions, mostly in the proteomic analysis part.

We are grateful for the reviewer’s positive criticism.

- Page 6, paragraph 3, related to figure 1G: The description of the dynamic range might be wrong. The dynamic range, as a ratio of the maximal and minimal intensity measured is 7 on a log₁₀ scale, which I believe is ~10⁷ fold, not ~7000 fold. Please, double check if this is correct.

We would like to thank the reviewer for bringing this to our attention. The calculation of dynamic range stated on page 6 is indeed faulty. We have now redone the calculations accordingly. There are two ways of reporting the ratio between quantifiable proteins in our experiment: 1) On the basis of the abundances of these proteins as detected in our UBIMAX experiment, indicating the measurable dynamic range with the UBIMAX method. 2) On the basis of the abundances of these same proteins but as detected in our total extract proteome, indicating the dynamic range of proteins for which ubiquitylation is detectable and quantifiable by UBIMAX. These numbers are 5.16E+04 (~50.000) and 3.76E+07 (~37.000.000), respectively.

We have changed this in the revised manuscript (page 6, 3rd paragraph), in which we have reported the latter:

“Overall, we observed a large dynamic range spanning seven orders of magnitude (3.76×10^7 -fold) with a distribution of ubiquitylated proteins identified by UBIMAX similar to that of a total egg extract proteome ($\text{Log}_{10} M = 7.68$ and 7.00 , respectively) (Fig. 1g and Supplementary Data 2), indicating a relatively modest abundance bias for UBIMAX.”

- Page 7, paragraph 1, related to figure 1H: The text mentions that the 786 proteins were further interrogated, and 4 clusters were identified and shown in the heatmap. However, the heatmap already shows a subset of these 786 proteins. It should be clear in the text what are the proteins shown in the heatmap used for the clustering.

We agree with the reviewer that the formulation is confusing, and have changed this sentence accordingly in the revised manuscript such that it now reads (page 7, 1st paragraph):

“From quadruplicate experiments, UBIMAX identified 1526 proteins of which 786 were significantly enriched by the His-pulldown and de novo ubiquitylated across the ubiquitin target enriched sample groups (Supplementary Fig. 1h, left and Supplementary Data 1a). We further interrogated a subset of these 786 proteins whose ubiquitylation status was consistently up- or downregulated across replicates and identified four clusters of specifically regulated ubiquitylated proteins in response to the DNA treatments (Fig. 1h).”

- Page 7, paragraph 1, related to figure 1H and S1H: The GO term enrichment analysis is based on a very small sample size which might give a significant result, but might be misleading. For example, figure S1H shows that the enrichment of the proteins related to inflammation in response to the addition of exogenous DNA was based on 2 proteins. The overrepresentation of the DNA repair proteins in the DSB-induced cluster seems to be more confident and based on more proteins. The authors might consider changing the description of their results in the part of describing the enrichment of the DNA-induced group or removing it. Furthermore, there are no terms related to inflammation in the Table S1, “Heatmap enrichment analysis” sheet, please double-check this supplementary file. Also consider matching the name of the clusters in this supplementary file to the heatmap and add a legend for this sheet to the Table S1 legend.

The reviewer has raised a relevant point regarding our enrichment analysis being based on a smaller sample size. We agree that the initially described analysis was not optimal and we have now redone it according to the reviewer’s suggestion. To this end, we have used a version of the UniProt *Xenopus laevis* database in which we have collapsed primary gene name (resulting in ~15,000 entries rather than the previous ~71,000) as background instead. Additionally, as the primary focus is indeed on the cluster of proteins whose ubiquitylation is upregulated in response to DSBs, we decided to focus the enrichment analysis on this cluster only and have accordingly removed the description of the enrichment in response to DNA in the text as the reviewer suggests. Reassuringly, this new analysis shows enrichment of terms related to DNA repair and DNA replication as upregulated in response to DSBs. As a result, we have replaced the data in old Supplementary Fig. 1h / new Supplementary Fig. 1i with the revised analysis as well as updated the supplementary data and -legend accordingly.

We also agree with the reviewer that visualization of intersection size in this figure could be confusing and have removed this aspect from old Supplementary Fig. 1h / new Supplementary Fig. 1i.

- Page 8, paragraph 1, related to figures 2D-I: If I understand it correctly, the data presented in the plots are based on two different MS experiments as shown in figure 1D and figure 2A. How did you analyze/process the intensity values from the two experiments, so the absolute values shown for DSB and ssDNA/SSB groups can be compared together?

The reviewer is correct that the abundance values presented in Fig. 2d-i originate from two different UBIMAX experiments. The two experiments were analysed separately in MaxQuant and in case a protein is detected in the same condition in both datasets (i.e. “no DNA” and “DNA”), the intensity values from both experiments (analyses) were added, without further processing, as separate data points in the graph. This is the reason for there being more than four data points for the no DNA and DNA conditions. As the variances between these data points are not significantly greater than within each of the two experiments (i.e. when looking at the DSB, ss-DNA-DPC and SSB-DPC conditions), we consider this data representation appropriate.

For the sake of clarity, we have labelled which data points originate from which UBIMAX experiment in the Source Data file.

- Page 12, paragraph 1, degron motif: Have you found this motif in other proteins enriched in your initial experiment, e.g., in Ku80?

The reviewer points to an interesting question. Indeed, we also performed an *in silico* analysis on the *Xenopus laevis* proteome and found several proteins exhibiting the variant β -Trcp1 degron, and which were also detected in our DSB-UBIMAX experiment (i.e. baz1a.L, dbn1.L, nadsyn1.L, nckipsd.S, psmd2.S, relch.S, sec31b.S, setd6, and tfrcl.L). However, of these, only Dbn1 showed regulation of ubiquitylation upon DSBs.

- Page 17, last paragraph: I think it would be more logical to include the in-silico motif analysis in the end of the results section rather than in the discussion.

We agree with the reviewer’s suggestion and have moved the paragraph describing the *in silico* degron analysis to the end of the last results section titled *The Dbn1 degron is a DDR-sensitive β -Trcp1 variant degron*.

- Page 38, Supplementary tables: Table S1 and Table S4 are not labelled in the sheets, and it was a bit difficult to understand which table is which. Furthermore, some of the supplementary tables are not called in the results section.

We thank the reviewer for their diligence in checking for these details in our manuscript. We have corrected the supplementary data labelling in both the tables themselves and their legends and inserted the missing callings in the revised manuscript.

Reviewer #3 (Remarks to the Author):

In this manuscript by Colding-Christensen et al, a protein ubiquitination profiling method by mass

spectrometry in *Xenopus* egg extract (names UBIMAX) is presented. The method relies on the addition of recombinant His-tagged ubiquitin to *Xenopus* egg extracts and the purification of ubiquitinated proteins under denaturing conditions, which ensures that ubiquitinated proteins are captured directly (i.e., ubiquitinated-protein to protein interactions are disrupted). As proof-of-concept, the authors focus on studying the DNA damage response when linearized plasmid DNA is added to *Xenopus* egg extracts (to simulate DNA double strand break (DSB) repair). This revealed several ubiquitinated proteins, including known DSB repair factors (e.g., Mre11, Nbn1, Rad50) and Dbn1, a protein that has not yet been implicated in DSB repair. The authors continue to characterise the mechanism of Dbn1 degradation and elegantly show that the cullin ring ligase (CRL) Cul1-beta-TRCP1 is responsible for degrading Dbn1 in response to DSBs. In the last part of the manuscript, the authors demonstrate that the degradation of Dbn1 is mediated by ATM and they identify the phosphor-degron.

The manuscript is well written and follows a clear flow of thoughts. The figures are informative and presented clearly. The experiments are well designed and include the correct controls. I recommend publication of the present manuscript and I have a few suggestions for the authors on how to further strengthen it.

We thank the reviewer for the kind words, and we are delighted that the reviewer finds our manuscript appropriate for publication in *Nature Communications*.

The authors perform all experiments by adding exogenous His-tagged ubiquitin to the *Xenopus* egg extracts. The authors include several controls to demonstrate that addition of recombinant ubiquitin does not induce any apparent artificial ubiquitination of target proteins. Nevertheless, a nice addition to this work would be to perform diGly peptide profiling as an orthogonal MS approach for quantifying ubiquitination sites in an endogenous setup.

We are happy that we have efficiently conveyed the relevance of performing these controls and sufficiently shown their value for reliably detecting ubiquitylated proteins by UBIMAX. We agree with the reviewer that endogenous ubiquitin site profiling in *Xenopus* egg extract would be a very nice additional approach. However, we would respectfully like to point out that a major practical consideration is the amount of starting material required for the diGly IP-MS approach. As described in the hallmark paper by Kim et al. in *Mol. Cell*, 2011, the diGly-IP-MS approach calls for 25-35 mg of protein input per immunoprecipitation sample, which would correspond to 700 μ L high speed supernatant (HSS) *Xenopus* egg extract. Hence, to perform a diGly-IP-MS setup orthogonal to the one introduced in our manuscript, would require egg extracts from 40 frogs. In contrast, for our UBIMAX analyses we use 100-fold less starting material (300 micrograms of protein per sample), meaning that the entire experiment outlined in Fig. 1d of our manuscript can be performed using eggs laid by 1 frog only.

Hence, while the orthogonal diGly IP-MS experiment may be theoretically feasible, it would be impractical considering the number of frogs needed, including ethical protocols for how frequently *Xenopus* frogs can be induced to lay eggs and, as an extension, the necessary use of experimental animals, as well as the maximum capacity of the instrumentation used for preparation of extract.

Due to these considerations, we have decided not to perform the diGly IP-MS experiments as suggested by the reviewer.

The authors convincingly demonstrate that beta-TRCP1 is the Cul1 substrate receptor responsible for degrading Dbn1 in response to DSBs. I am wondering whether the authors investigated downstream

consequences of Dbn1 depletion (or introduction of the non-degradable ATM phosphor-deficient mutant)? For example, how is DSB repair affected when Dbn1 is depleted, or if cells cannot degrade Dbn1 in response to DSBs? Is cancer cell survival affected by non-degradable (or hyper-phosphorylated) Dbn1 mutants? Since Dbn1 is an actin cytoskeleton organizing protein, are there consequences for cell division when Dbn1 is depleted? And are there any known cancer mutations in either Dbn1 or beta-TRCP1? It would be helpful, if the authors could discuss these points.

We thank the reviewer for these insightful comments, and we are delighted that the reviewer finds that we have convincingly shown the molecular mechanism for DSB-induced Dbn1 ubiquitylation.

We are also thankful for their very useful suggestions on how to further investigate the functional consequences of Dbn1 status on DSB repair. We have attempted to answer each of the reviewer's questions by various experimental strategies.

The first of the reviewer's suggestions were also asked by reviewer 1 (point #5) and we refer to the discussion and data described here. In brief, we did not observe any consistent effects on DSB repair, DDR signalling or cancer cell survival with either depletion of Dbn1 or reconstitution with the non-degradable Dbn1 degenon mutant in either *Xenopus* egg extracts or in human cells.

The reviewer's suggestion that DBN1 status could have consequences for cell division through its regulation of the actin cytoskeleton is very interesting. However, our Incucyte experiments performed in the absence of DNA damage (Rebuttal Fig. 11) indicate that DBN1 KO and DBN1-S599A cells have no growth defect when compared to parental/WT cells.

Rebuttal Fig. 11. Incucyte assay measuring cell confluency with time in either parental, CRISPR *DBN1* KO, or the latter complemented with stable inducible expression of WT or *DBN1-S599A* mutant in unperturbed conditions. This experiment was done in technical (cell culture) triplicates. Displayed here are the mean values with error bars representing standard deviations.

To test whether there is a difference in the cell cycle distribution in these same cell lines, we performed a ScanR microscopy experiment and quantified the proportions of cells in G1-, S- and G2/M-phases by quantifying and correlating staining with DAPI and for PCNA foci (Rebuttal Fig. 12). Here, we also found no apparent differences with either loss of *DBN1* or gain of the stable *DBN1-S599A* mutant. Together, these data suggest that *DBN1* status does not affect general cell division.

Rebuttal Fig. 12. Human cells either parental, CRISPR *DBN1* KO, or the latter complemented with stable inducible expression of WT or *DBN1*-S599A mutant were assayed for cell cycle distribution based on fluorescence microscopy analysis of PCNA foci correlated with DAPI staining.

With regards to any known cancer mutations of the genes encoding *DBN1* and β -TRCP1, we searched the NIH national cancer institute CDC data portal. According to this database, there are 153 reported cases of *DBN1* mutation in cancer. This is distributed between 150 different mutations in the gene of which there is one case of a substitution, S599F, within the degron. However, this particular mutation is reported to have a moderate/benign effect on the transcript and protein sequence. For the *BTRC* gene, encoding β -TRCP1, there are 142 cases reported with 131 different mutations. In 18 of these cases, patients had at least one mutation in both genes.

We suspect that the lack of an effect in the experiments described above could, at least in part, be due to only a specific pool of *DBN1* protein being targeted for ubiquitylation upon DSBs (i.e. nuclear *DBN1* in G1 phase cells). It may be necessary (but quite experimentally complicated) to target this very specific pool of *DBN1* to see a clear effect of *DBN1* ubiquitylation status on DSB repair, cell cycle and -survival.

Figure 1E: Besides showing quantitative correlations, it would be interesting to see how precise the quantification of the detected proteins is.

We thank the reviewer for this suggestion and agree that analysing the coefficients of variation (CVs) within the DSB-UBIMAX experiment is a good measure for how precisely UBIMAX quantifies ubiquitylated proteins. Hence, following the reviewer's suggestion we have performed this analysis within the five conditions outlined in Fig. 1d. We find that the distribution of CVs is similar for the five conditions and that the median CVs are low, supporting that quantification of ubiquitylated proteins is accurate with our UBIMAX method. We have added this figure to the revised manuscript (new Supplementary Fig. 1d).

Figure S1G: Indicate how many proteins were identified in how many conditions (to get a better understanding on the data completeness/number of missing values)

We are uncertain about the specific request made by the reviewer, but presume the reviewer is seeking information about data completeness and missing values across conditions and replicates. To address the latter, we have summarized how many proteins are detected in one, two, three or all four replicates within each condition as well as in the experiment as a whole (in this case then in one, two, three or all four replicates in at least one of the five conditions, Rebuttal Fig. 13a). To address the former, we have produced an upset plot (Rebuttal Fig. 13b) indicating how many of the proteins that are detected in all four replicates in at least one of the five conditions (1526 proteins) are detected in all four replicates across the five different conditions of the DSB-UBIMAX experiment.

Rebuttal Fig. 13. a. Number of proteins detected in one, two, three or four replicates across the five conditions of the DSB-UBIMAX experiment or the experiment as a whole (Total exp). **b.** Upset plot showing the number of proteins detected exclusively or in two, three, four or all five conditions of the DSB-UBIMAX experiment.

We do not believe that this data would add significant additional value to the manuscript and have therefore opted to not include it in the revised manuscript. But we are of course happy to reconsider this if the reviewer thinks differently.

Figure 3C: The proof that dominant negative Cul1 supplementation to the egg extracts rescues Dbn1 degradation looks convincing; however, I noticed that also Cul4a seems to rescue its degradation. Could the authors comment on this? Is it possible that more than one CRL complex regulate Dbn1 protein stability?

We agree that from the experiment shown in Fig. 3d, we cannot exclude a contribution from Cul4a ligase complexes on Dbn1 ubiquitylation. This point was also raised by reviewer 1 (other point #2) and we refer to the discussion and data presented in connection with this point. In brief, we have since realised that the initial experiment unfortunately was performed with varying amounts of Cullin dominant negative proteins. We have therefore redone the experiment with titrated amounts of Cullin dominant negative proteins and replaced the panel in Fig. 3d in the revised manuscript. We believe that the new data fully supports our initial conclusion and should alleviate the concerns raised by the reviewer.

References

1. Akimov, V. et al. UbiSite approach for comprehensive mapping of lysine and N-terminal ubiquitination sites. *Nat Struct Mol Biol* **25**, 631-640 (2018).
2. Kim, W. et al. Systematic and quantitative assessment of the ubiquitin-modified proteome. *Mol Cell* **44**, 325-40 (2011).
3. Hendriks, I.A., Akimov, V., Blagoev, B. & Nielsen, M.L. MaxQuant.Live Enables Enhanced Selectivity and Identification of Peptides Modified by Endogenous SUMO and Ubiquitin. *J Proteome Res* **20**, 2042-2055 (2021).
4. Mellacheruvu, D. et al. The CRAPome: a contaminant repository for affinity purification-mass spectrometry data. *Nat Methods* **10**, 730-6 (2013).
5. Long, D.T., Raschle, M., Joukov, V. & Walter, J.C. Mechanism of RAD51-dependent DNA interstrand cross-link repair. *Science* **333**, 84-7 (2011).
6. Aymard, F. et al. Genome-wide mapping of long-range contacts unveils clustering of DNA double-strand breaks at damaged active genes. *Nat Struct Mol Biol* **24**, 353-361 (2017).
7. Belin, B.J., Lee, T. & Mullins, R.D. DNA damage induces nuclear actin filament assembly by Formin -2 and Spire-(1/2) that promotes efficient DNA repair. [corrected]. *Elife* **4**, e07735 (2015).
8. Caridi, C.P. et al. Nuclear F-actin and myosins drive relocalization of heterochromatic breaks. *Nature* **559**, 54-60 (2018).
9. Schrank, B.R. et al. Nuclear ARP2/3 drives DNA break clustering for homology-directed repair. *Nature* **559**, 61-66 (2018).
10. Zagelbaum, J. et al. Multiscale reorganization of the genome following DNA damage facilitates chromosome translocations via nuclear actin polymerization. *Nat Struct Mol Biol* **30**, 99-106 (2023).
11. Larsen, N.B. et al. Replication-Coupled DNA-Protein Crosslink Repair by SPRTN and the Proteasome in Xenopus Egg Extracts. *Mol Cell* **73**, 574-588 e7 (2019).
12. Postow, L. & Funabiki, H. An SCF complex containing Fbx12 mediates DNA damage-induced Ku80 ubiquitylation. *Cell Cycle* **12**, 587-95 (2013).
13. Postow, L. et al. Ku80 removal from DNA through double strand break-induced ubiquitylation. *J Cell Biol* **182**, 467-79 (2008).
14. Frescas, D. & Pagano, M. Deregulated proteolysis by the F-box proteins SKP2 and beta-TrCP: tipping the scales of cancer. *Nat Rev Cancer* **8**, 438-49 (2008).
15. Margottin, F. et al. A novel human WD protein, h-beta TrCp, that interacts with HIV-1 Vpu connects CD4 to the ER degradation pathway through an F-box motif. *Mol Cell* **1**, 565-74 (1998).

Reviewer #1 (Remarks to the Author):

I appreciate the authors' rigor in addressing my points. My main concern is still that the manuscript is written as a proteomic technique paper but does not make a substantial proteomic advance. It uses an old technique of tagged ubiquitin pulldown, which is limited in ubiquitylated protein identification, as the authors acknowledge, and does not examine endogenous ubiquitylation. The technique is not new to *Xenopus* oocyte extracts, having been performed previously for another ubiquitin-like protein (Ma, Mol Cell Proteomics 2014)

That said, the proteomics are well controlled, and their finding that Dbn1 is ubiquitylated in response to DSBs by Cul1- β TRCP is interesting. In my mind, their Dbn1 results are a greater contribution than the proteomics. I support publication if they include their functional data on Dbn1 in DNA repair. I realize these findings are negative, but the field will be interested in these experiments, which are a natural and important extension of their work. Their Discussion (p. 17) says it would be interesting to understand if Dbn1 degradation impacts DSB repair, and they have already performed initial crucial experiments in answering this question.

1. The authors demonstrate nicely that Dbn1 depletion (and addback of the ubiquitination resistant Dbn1 mutant) does not affect NHEJ (Rebuttal Fig 3) or DNA end resection (Rebuttal Fig 4) in *Xenopus* extracts. Nor does it affect homologous recombination (Rebuttal Fig 5), H2AX foci resolution (Rebuttal Fig 6), or DSB sensitivity (Rebuttal Figs 7-8) in mammalian cells. These data should be included in the manuscript. The results are negative, but their suggestion that Dbn1 may regulate DSB repair more subtly through regulation of DSB movement is reasonable.

2. The authors acknowledge a high level of non-specific protein binding in their proteomic data, (my prior points 2, 3 and other point 1). They correctly show that that non-specific binding is also prevalent in diGly and UbiSite approaches (see their response to my point 2). However, these latter approaches only report ubiquitylated peptides (i.e., diGly containing peptides) in proteomic tables, whereas their proteomic tables include peptides from non-specific binding proteins. This makes proper controls (i.e., no His, E1 inhibitor) important, which is a strength of their paper. They should directly acknowledge in their text a high degree of non-specific protein enrichment, which necessitates their no His and E1 inhibitor controls. I see this as an asset of their control design. Readers should recognize the importance of using these controls to confidently identify ubiquitylated proteins in their UBIMAX proteomic tables (i.e., Tables S1, S3).

3. They should include their proteasome inhibitor UBIMAX experiments (Rebuttal Fig 2) in the manuscript. It is well known that proteasome inhibition can have indirect effects on ubiquitin dynamics by causing ubiquitin pool depletion and can thereby prevent non-degradative ubiquitylation events. The reader should have this information to help determine which ubiquitylation events may be degradative.

4. In response to my point 9, the authors show that summed peptide intensity is the same across samples. This increases confidence that protein input is the same for the samples. However, the authors have still not addressed why ATMi inhibits so many Dbn1 interactions. As shown in Table S4, the DSB / DSB + ATMi sample has only 18 proteins with an ATMi-induced 1.5-fold increase in Dbn1 interaction ($\text{Log}_2 < -0.6$) yet 75 proteins with a 1.5-fold decrease in interaction ($\text{Log}_2 > 0.6$) among a total of 231 proteins.

5. They should cite Ma et al. Mol Cell Prot 2014 more clearly as a tagged ubiquitin-like approach that was previously used in *Xenopus* extracts.

Reviewer #2 (Remarks to the Author):

The authors addressed my comments sufficiently.

Reviewer #3 (Remarks to the Author):

The authors have addressed all my concerns adequately and I recommend publication of this work. I agree with not including an additional experiment based on ubiquitin remnant profiling and the upset plot shown in Rebuttal Fig. 13.

I do, however, have a small additional note for the authors. The cited references on diGly peptide profiling are quite outdated (e.g., Kim et al, 2011) and newer approaches based on DIA-MS require much less protein input (microgram range), have a higher quantification precision and a better reproducibility (e.g., recent work from the Mann lab, <https://doi.org/10.1038/s41467-020-20509-1>).

Congratulations to all authors!

REVIEWERS' COMMENTS

We are grateful to all three reviewers for endorsing publication of our manuscript in Nature Communications. Similarly, we are happy that all three reviewers express that we have sufficiently addressed their comments. Below, we have responded to the final comments of the reviewers, if any, and detailed the final changes made to our manuscript based on these.

Reviewer #1 (Remarks to the Author):

I appreciate the authors' rigor in addressing my points. My main concern is still that the manuscript is written as a proteomic technique paper but does not make a substantial proteomic advance. It uses an old technique of tagged ubiquitin pulldown, which is limited in ubiquitylated protein identification, as the authors acknowledge, and does not examine endogenous ubiquitylation. The technique is not new to *Xenopus* oocyte extracts, having been performed previously for another ubiquitin-like protein (Ma, Mol Cell Proteomics 2014)

That said, the proteomics are well controlled, and their finding that Dbn1 is ubiquitylated in response to DSBs by Cul1-TRCP is interesting. In my mind, their Dbn1 results are a greater contribution than the proteomics. I support publication if they include their functional data on Dbn1 in DNA repair. I realize these findings are negative, but the field will be interested in these experiments, which are a natural and important extension of their work. Their Discussion (p. 17) says it would be interesting to understand if Dbn1 degradation impacts DSB repair, and they have already performed initial crucial experiments in answering this question.

1. The authors demonstrate nicely that Dbn1 depletion (and addback of the ubiquitination resistant Dbn1 mutant) does not affect NHEJ (Rebuttal Fig 3) or DNA end resection (Rebuttal Fig 4) in *Xenopus* extracts. Nor does it affect homologous recombination (Rebuttal Fig 5), H2AX foci resolution (Rebuttal Fig 6), or DSB sensitivity (Rebuttal Figs 7-8) in mammalian cells. These data should be included in the manuscript. The results are negative, but their suggestion that Dbn1 may regulate DSB repair more subtly through regulation of DSB movement is reasonable.

We appreciate that the reviewer acknowledges our substantial work in attempting to reveal the functional significance of DSB-induced ubiquitylation of Dbn1 and we are happy to let the above findings, though negative, be available to the field through the rebuttal document published along with this manuscript.

2. The authors acknowledge a high level of non-specific protein binding in their proteomic data, (my prior points 2, 3 and other point 1). They correctly show that that non-specific binding is also prevalent in diGly and UbiSite approaches (see their response to my point 2). However, these latter approaches only report

ubiquitylated peptides (i.e., diGly containing peptides) in proteomic tables, whereas their proteomic tables include peptides from non-specific binding proteins. This makes proper controls (i.e., no His, E1 inhibitor) important, which is a strength of their paper. They should directly acknowledge in their text a high degree of non-specific protein enrichment, which necessitates their no His and E1 inhibitor controls. I see this as an asset of their control design. Readers should recognize the importance of using these controls to confidently identify ubiquitylated proteins in their UBIMAX proteomic tables (i.e., Tables S1, S3).

We are happy that the reviewer agrees with and acknowledges our approach in rigorously controlling for background-binding proteins through our chosen experimental design. We detail this control design and the reason for employing it in the results section (starting on page 5, last paragraph) of our manuscript. The statistical data underlying the separation of background-binding and ubiquitylated proteins is available in its entirety in Supplementary Data 1 and 3 for the DSB- and DPC-UBIMAX experiments, respectively.

3. They should include their proteasome inhibitor UBIMAX experiments (Rebuttal Fig 2) in the manuscript. It is well known that proteasome inhibition can have indirect effects on ubiquitin dynamics by causing ubiquitin pool depletion and can thereby prevent non-degradative ubiquitylation events. The reader should have this information to help determine which ubiquitylation events may be degradative.

We agree with the reviewer that this data is useful for further considering the degradative potential of DSB-induced ubiquitylation as well as for visualising the wide use of UBIMAX in investigating protein ubiquitylation under highly precise conditions of interest and we are happy to publicise this data in the rebuttal document for readers of this manuscript.

4. In response to my point 9, the authors show that summed peptide intensity is the same across samples. This increases confidence that protein input is the same for the samples. However, the authors have still not addressed why ATMi inhibits so many Dbn1 interactions. As shown in Table S4, the DSB / DSB + ATMi sample has only 18 proteins with an ATMi-induced 1.5-fold increase in Dbn1 interaction ($\text{Log}_2 < -0.6$) yet 75 proteins with a 1.5-fold decrease in interaction ($\text{Log}_2 > 0.6$) among a total of 231 proteins.

We agree with the reviewer that the potential ATM-mediated regulation of the Dbn1 interaction landscape is intriguing. However, we would like to point out that of the 18 and 75 proteins mentioned by the reviewer, three show a statistically significant increase, while 22 proteins show a statistically significant decrease in interaction with Dbn1 upon ATM inhibition, respectively.

Previous work has shown that actin filament polymerization required for movement and efficient repair of DSBs by homologous recombination in S/G2 phase of the cell cycle, depends on ATM activity (Schrank et al., Nature, 2018). In the present manuscript, we show that ATM activity is required for DSB-induced phosphorylation and ubiquitylation of Dbn1. With Dbn1 being an actin-organizing protein, it is tempting to speculate that there may also be a connection between the ATM-mediated regulation of Dbn1 phosphorylation, ubiquitylation and stability and actin filament dynamics in response to DSBs. Such a regulation could indeed be accomplished, at least in part, through phosphorylation and ubiquitylation-mediated modification of Dbn1, actin, and other actin-related factors, resulting in a change of the interactions between these proteins. This could indeed be one interpretation of our Dbn1 IP-MS data. However, as we, in this experiment, cannot distinguish between proteins interacting with phosphorylated, ubiquitylated and unmodified Dbn1, we are not comfortable making such speculations in our manuscript.

5. They should cite Ma et al. Mol Cell Prot 2014 more clearly as a tagged ubiquitin-like approach that was previously used in *Xenopus* extracts.

In response to the reviewer's request, we have moved the Ma et al., Mol Cell Prot, 2014 reference from where it was located before (page 3, 2nd paragraph) to another sentence (on page 3, 3rd paragraph), which now more clearly describes the reference:

"Recently, Xenopus egg extracts have been coupled to MS-based proteomic analyses to study quantifiable changes in protein recruitment to damaged DNA^{26,27} as well as for identifying small ubiquitin-like modifier substrates through a tagged protein approach²⁸."

Reviewer #2 (Remarks to the Author):

The authors addressed my comments sufficiently.

Reviewer #3 (Remarks to the Author):

The authors have addressed all my concerns adequately and I recommend publication of this work. I agree with not including an additional experiment based on ubiquitin remnant profiling and the upset plot shown in Rebuttal Fig. 13.

I do, however, have a small additional note for the authors. The cited references on diGly peptide profiling are quite outdated (e.g., Kim et al, 2011) and newer approaches based on DIA-MS require much less protein input (microgram range), have a higher quantification precision and a better reproducibility (e.g., recent work from the Mann lab, <https://doi.org/10.1038/s41467-020-20509-1>).

Congratulations to all authors!

We appreciate the reviewer's positive remarks on our manuscript. While we acknowledge the existing literature on employing DIA-MS for investigating ubiquitylation sites, we believe it is important to note that the authors of the referenced paper utilized a spectral library generated through standardized DDA-MS to conduct their lower-input material DIA-MS. Consequently, the applicability of the lower-input advancements in the DIA-MS approach is contingent upon a prior higher-input step. Furthermore, the DIA-MS method outlined by Hansen *et al.* is limited to detecting ubiquitylation events already defined in the spectral library, making it primarily suitable for high-throughput screening rather than uncovering novel insights into ubiquitylated substrates (i.e., a discovery approach) – a capability that UBIMAX offers.

Consequently, we have not cited the Hansen *et al.* paper in our manuscript, as the focus of the outlined DIA-MS strategy differs from the scientific scope of our UBIMAX method.